# No Free Lunch in Self Supervised Representation Learning

## Abstract

Self-supervised representation learning in computer vision relies heavily on hand-crafted image transformations to learn meaningful and invariant features. However few extensive explorations of the impact of transformation design have been conducted in the literature. In particular, although the dependence of representation quality to transformation design has been established, it has not been thoroughly studied. In this work, we explore this relationship and its impact on a domain other than natural images. We demonstrate that designing transformations can be viewed as a form of beneficial supervision. Firstly, we not only show that transformations have an effect on the features in representations and the relevance of clustering, but also that each category in a supervised dataset can be impacted differently in a controllable manner. Furthermore, we explore the impact of transformation design on a domain such as microscopy images where differences between classes are more subtle than in natural images. In this case, we observe a more significant impact on the features encoded into the resulting representations. Finally, we demonstrate that transformation design can be leveraged as a form of supervision, as careful selection of these transformation, based on the desired features, can lead to a drastic increase in performance by improving the resulting representation.

## 1 Introduction

In Self-Supervised Representation Learning (SSRL), a common learning objective in most approaches is for models to be trained to learn a common representation of two different transformations of the same image. The objective of SSRL is to benefit from training on a large unannotated dataset to obtain a representation that can be useful at solving downstream tasks for which one has a limited amount of annotated data. SSRL has become one of the main pillars of *Deep Learning* based computer vision approaches Bardes et al. (2022a); Caron et al. (2021; 2018); Chen et al. (2020a;b); Grill et al. (2020); Zbontar et al. (2021), with performances coming close to, and sometimes going beyond supervised learning for some downstream tasks.

SSRL relies heavily on combinations of random image transformations. These transformations are used to create distorted versions of the original image with the aim of keeping the semantic content invariant. With SSRL approaches producing overall good accuracies on downstream classification of natural images, hyper-parameter optimization of transformation parameters has added significant improvements to the overall performance of models Chen et al. (2020a). However, further consequences of the choice of these augmentations have been only sporadically explored by the research community Grill et al. (2020); Wagner et al. (2022), especially for other tasks Zhang & Ma (2022) and in other domains Xiao et al. (2021). It is therefore unclear to what extent this choice impacts the pretraining of models at deeper levels, as well as the effects on the extracted features and the performances in other domains.

Some important questions remain unanswered. Is the accuracy for an individual class contingent upon the choice of augmentation ? Can the variation of this choice increase one class accuracy at the expense of degrading another one? Are the features encoded into the latent representations being affected by this choice ? What is the amplitude of these issues in domains other than natural images?

In this paper, we report and analyze the outcomes of our experimentation to shed light on this subject. By examining the performance of various SSRL methods, while altering the selection and magnitude of

transformations, we analyze and quantify their ramifications on overall performance as well as on the class-level performance of models. We subsequently seek to observe the effects of substantially varying the structure of combinations of transformations on the quality of the resulting representations. We then investigate the effects of varying selections transformations in SSRL methods when applied to microscopy images of cells, where distinctions between classes are far less discernible than with natural images, and then proceed to discuss potential avenues for improvement in the SSRL field.

Using convolution based approaches on small to medium scale datasets, our contributions can be succinctly summarized as follows:

- We explore the nuanced impact of transformations on performance at the class level. Our investigation reveals that the selection of transformations commonly used in Self-Supervised Learning approaches can degrade the accuracy of certain classes while improving those of others. However, we also observe that this effect is controllable and can be leveraged to beneficially manage the performance of specific classes in specific application scenarios.

- We demonstrate that, through analysis of the representations obtained from Self-Supervised trainings, the careful selection of specific combinations of transformations facilitates the optimization of models for encoding distinct features into the resulting representations. Simultaneously, this deliberate choice may result in the loss of other features, thus enabling models to be tailored for different tasks.

- We examine the implications of the choice of transformations for Self-Supervised Learning in the biological domain, where distinction between classes is often fuzzy and subtle. Our findings illuminate the heightened importance of transformation choice within this domain, showcasing that a meticulous definition of desired features yields improvements in result quality. Moreover, our experiments demonstrate the superiority of this approach over transfer learning when dealing with small-scale datasets exhibiting domain differences.

## 2 Related work

**Self supervised representation learning (SSRL).** Contrastive learning approaches Chen et al. (2020a; 2021; 2020b); Yeh et al. (2022) have shown great success in avoiding trivial solutions in which all representations collapse into a point, by pushing the original image representation further away from representations of negative examples. These approaches follow the assumption that the augmentation distribution for each image has minimal inter-class overlap and significant intra-class overlap Abnar et al. (2022); Saunshi et al. (2019). This dependence on contrastive examples has since been bypassed by non contrastive methods. The latter either have specially designed architectures Caron et al. (2021); Chen & He (2021); Grill et al. (2020) or use regularization methods to constrain the representation in order to avoid the usage of negative examples Bardes et al. (2022a); Ermolov et al. (2021); Lee & Aune (2021); Li et al. (2022); Zbontar et al. (2021); Bardes et al. (2022b). Another line of work Kalantidis et al. (2020); Shah et al. (2023) focuses on obtaining positive and negative examples in the feature space, bypassing the need of augmenting the input images with transformations.

**Impact of image transformations on SSRL.** Compared to the supervised learning field, the choice and amplitude of transformations has not received much attention in the SSRL field Balestriero et al. (2022); Cubuk et al. (2019); Li et al. (2020); Liu et al. (2021). Studies such as Wen & Li (2021) and Wang & Isola (2020) analyzed in a more formal setting the manner in which augmentations decouple spurious features from dense noise in SSRL. Some works Chen et al. (2020a); Geirhos et al. (2020); Grill et al. (2020); Perakis et al. (2021) explored the effects of removing transformations on the overall accuracy. Other works explored the effects of transformations by capturing information across each possible individual augmentation, and then merging the resulting latent spaces Xiao et al. (2021), while some others suggested predicting intensities of individual augmentations in a semi-supervised context Ruppli et al. (2022). However the latter approach is limited in practice as individual transformations taken alone were shown to be far less efficient than compositions Chen et al. (2020a). An attempt was made to explore the underlying effect of the choice of transformation in the work of Pal et al. (2020), one of the first works to discuss how certain transformations

are better adapted to some pretext task in self supervised learning. This study suggests that the best choice of transformations is a composition that distorts images enough so that they are different from all other images in the dataset. However favoring transformations that learn features specific to each image in the dataset should also degrade information shared by several images in a class, thus damaging model performance. Altogether, it seems that a good transformation distribution should maximize the intra-class variance, while minimizing inter-class overlap Abnar et al. (2022); Saunshi et al. (2019). Other works proposed a formalization to generalize the composition of transformations Patrick et al. (2021), which, while not flexible, provided initial guidance to improve results in some contexts. This was followed by more recent works on the theoretical aspects of transformations, von Kügelgen et al. (2021) that studied how SSRL with data augmentations identifies the invariant content partition of the representation, Huang et al. (2021) that seeks to understand how image transformations improve the generalization aspect of SSRL methods, and Zhang & Ma (2022) that proposes new hierarchical methods aiming to mitigate a few of the biases induced by the choice of transformations.

**Learning transformations for SSRL.** A few studies showed that optimizing the transformation parameters can lead to a slight improvement in the overall performance in a low data annotation regime Ruppli et al. (2022); Reed et al. (2021). However, the demonstration is made for a specific downstream task that was known at SSRL training time, and optimal transformation parameters selected this way were shown not to be robust to slight change in architecture or task Saunshi et al. (2022). Other works proposed optimizing the random sampling of augmentations by representing them as discrete groups, disregarding their amplitude Wagner et al. (2022), or through the retrieval of strongly augmented queries from a pool of instances Wang & Qi (2021). Further research aimed to train a generative network to learn the distribution of transformation in the dataset through image-to-image translation, in order to then avoid these transformations at self supervised training time Yang et al. (2021). However, this type of optimization may easily collapse into trivial transformations.

**Performance of SSRL on various domains and tasks.** Evaluation of SSRL works relies almost exclusively on the accuracy of classification of natural images found in widely used datasets such as Cifar Krizhevsky (2009), Imagenet Deng et al. (2009) or STL Coates et al. (2011). This choice is largely motivated by the relative ease of interpretation and understanding of the results, as natural images can often be easily classified by eye. This, however, made these approaches hold potential biases concerning the type of data and tasks for which they could be efficiently used. It probably also has an impact on the choice and complexity of the selected transformations aiming at invariance: some transformations could manually be selected in natural images but this selection can be very challenging in domains where differences between classes are invisible. The latter was intuitively mentioned in some of the previously cited studies. Furthermore, the effect of the choice of transformation may be stronger on domains and task where the representation is more thoroughly challenged. This is probably the case in botany and ornithology Xiao et al. (2021) but also in the medical domain Ruppli et al. (2022) or research in biology Bourou et al. (2023); Lamiable et al. (2022); Masud et al. (2022).

## 3 The choice of transformations is a subtle layer of weak supervision

In Section 3.1, we empirically investigate the ramifications of varying transformation intensities on the class-level accuracies of models trained using self-supervised learning techniques. Subsequently, in Section 3.2, we conduct an examination in which we demonstrate how alternative selections of transformations can lead to the optimization of the resulting representations of the model for distinct use cases. In Section 3.3, we delve into the manner in which this choice can impact representations of microscopy images, a domain where distinction between images is highly nuanced. This is followed by an empirical analysis in Section 3.4 that illustrates how the combination of transformations chosen according to a meticulous definition of desired biological features can significantly enhance the performance of models in SSRL.

### 3.1 Transformation choices induce inter-class bias

In order to understand the ramifications of transformations on the performance of a model, we delve into the examination of the behavior of models that are trained with widely adopted SSRL techniques on the benchmark datasets Cifar10, Cifar100 Krizhevsky (2009) and Imagenet100 Deng et al. (2009), while altering

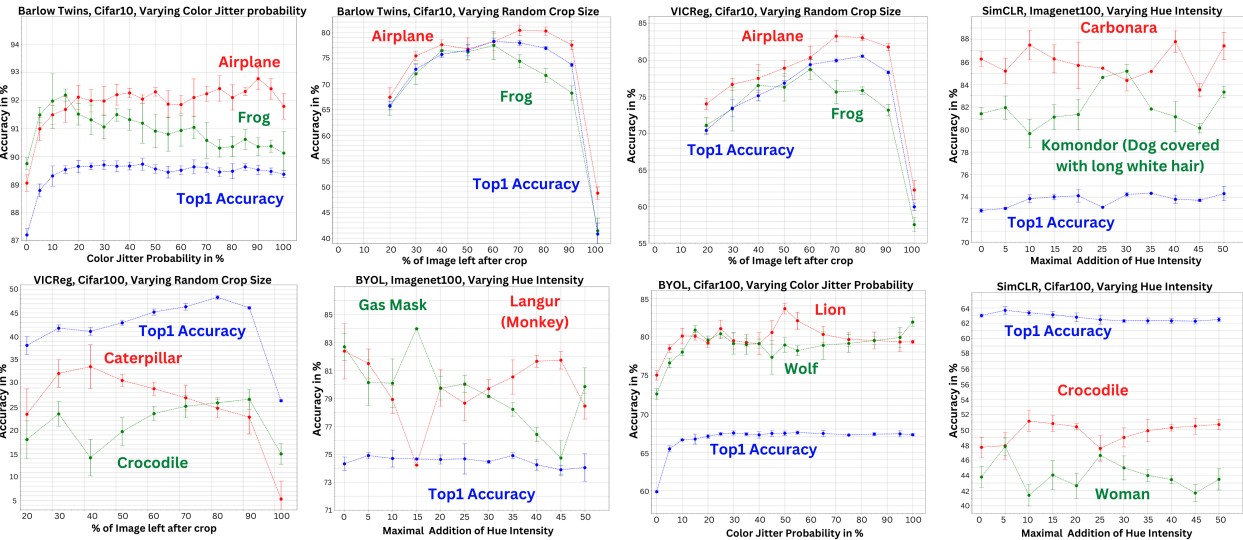

Figure 1: **Different transformation parameter choices induce an inter-class bias.** Inter-class accuracy results versus variation of a transformation parameter, for Resnet18 architectures trained with various SSRL methods on the benchmark datasets Cifar10, Cifar100 and Imagenet100. Each dot and associated error bar reflects the mean and standard deviation of three runs for Imagenet100 and five runs for Cifar with different random seeds. While overall accuracy remains relatively consistent across a range of transformation parameters, these transformations can have a subtle but significant impact on individual class performance, either favoring or penalizing specific classes. Additional comparisons are available in Supplementary Materials, Figure 11.

the magnitude and likelihood of the transformations. With a Resnet18, a Resnet50 and a ConvNeXt-Tiny architectures as backbones, we employ a fixed set of transformations, comprised of randomized cropping, chromatic perturbations, and randomized horizontal inversions. Subsequently, we uniformly sample a set of amplitude and probability values for each transformation, in order to create a diverse range of test conditions. Each training is repeated a number of times (three for Imagenet and five for Cifar), with distinct seed values, and the mean and standard deviation of accuracy, measured through linear evaluation over frozen weights, are computed over these five trainings for each method and each transformation value. All model training parameters, as well as the training process, are available in Supplementary Materials B.1.

As depicted in Figure 1, we observe minimal fluctuation in the overall accuracy of each model as we slightly alter any one of the transformations. This stands in stark contrast to the class-level accuracies observed, in which we discern significant variation in the accuracy value for many classes, as we vary the parameters of transformations, hinting at a greater impact of variations in transformation parameters on the class-level. Through the same figure, it becomes apparent that a number of classes exhibit distinct, and at times, entirely antithetical behaviors to each other within certain ranges of a transformation parameter. In the context of the datasets under scrutiny, this engenders a bias in the conventional training process of models, which either randomly samples transformation parameters or relies on hyperparameter optimization on overall accuracy to determine optimal parameters. This bias manifests itself in the manner in which choosing specific transformation parameters would impose a penalty on certain classes while favoring others. This is demonstrated in Figure 1 by the variation in accuracy of the Caterpillar and Crocodile classes for a model trained using VICReg Bardes et al. (2022a), as the crop size is varied (bottom left plot). The reported accuracies uncover that smaller crop sizes prove advantageous for the Caterpillar class, stimulating the model to recognize repetitive patterns and features consistent across the length of the caterpillar's body. However, the Crocodile class doesn't fare as well under similar conditions. This can be explained by considering the differing morphologies of the two subjects. The Caterpillar class benefits from smaller crops as the caterpillars exhibit uniformity across their body parts. Conversely, for the Crocodile class, a small crop size could

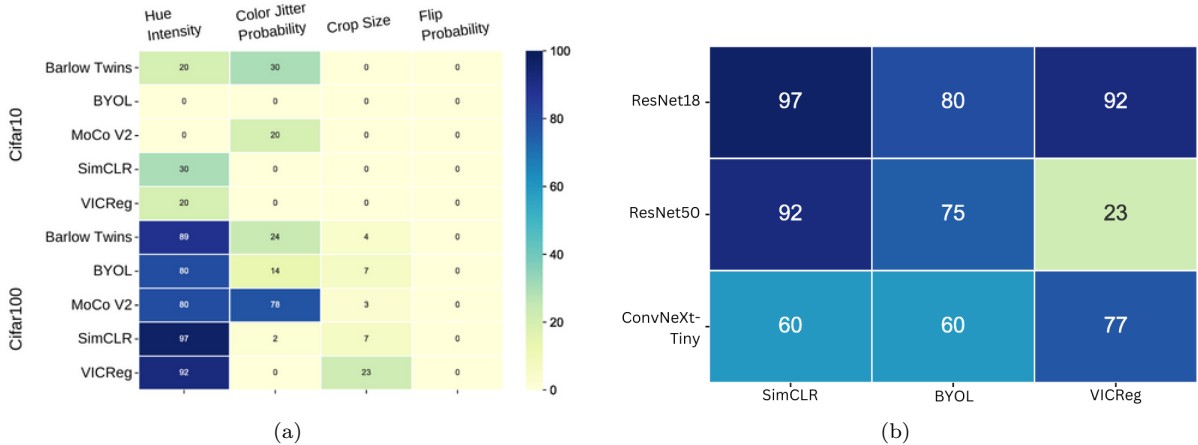

(a)                    (b)

Figure 2: **A comprehensive analysis to quantify the occurrences of negative correlations among individual classes accuracies in the Cifar10 and Cifar100 datasets.** This analysis is conducted using different backbones and SSRL approaches, along with various transformations, through examination of the ratio of classes exhibiting negative correlations with each other in relation to the total number of classes within Cifar10 and Cifar100. (a) Notably, within Cifar100 with ResNet18, a higher proportion of negatively correlated classes is observed, which can be attributed to the increased number of classes that may overlap when increasing color jitter. (b) We employ different backbones (ResNet18, ResNet50, and ConvNeXt-Tiny) and self-supervised approaches (SimCLR, BYOL, and VICReg) on Cifar100 while manipulating the hue intensity. We compute the ratio of classes exhibiting negative correlations with each other in relation to the total number of classes within Cifar100. The results obtained from these various configurations remain in similar ranges of each other while varying both the SSRL approach and the backbone used. This provides an evidence that the observed patterns in (a) are independant of the SSL method and encoder architectures.

potentially capture a segment like the tail, which could be misattributed to other classes, such as snakes, due to its isolated resemblance. Therefore, the choice of transformation probability or intensity directly affects class-level accuracies, an impact that may not be immediately apparent when only considering the overall accuracy.

In order to gain deeper understanding of the inter-class bias observed in our previous analysis, we aim to further investigate the extent to which this phenomenon impacts the performance of models trained with self-supervised learning techniques. By quantitatively assessing the correlation scores between class-level accuracies obtained under different transformation parameters, we aim to measure the prevalence of this bias in self-supervised learning methods. More specifically, a negative correlation score between the accuracy of two classes in response to varying a given transformation would indicate opposing reactions for those classes to the transformation parameter variations. Despite its limitations, such as the inability to quantify the extent of bias and the potential for bias to manifest in specific ranges while remaining positively correlated in others (See Lion/Wolf pair in Figure 1), making it difficult to detect, this measure can still provide a preliminary understanding of the degree of inter-class bias. To this end, we conduct a series of experiments utilizing a ResNet18 encoder on the benchmark datasets of Cifar10 and Cifar100 Krizhevsky (2009). We employ a diverse set of state-of-the-art self-supervised approaches: Barlow Twins Zbontar et al. (2021), MoCov2 Chen et al. (2020b), BYOL Grill et al. (2020), SimCLR Chen et al. (2020a), and VICReg Bardes et al. (2022a), and use the same fixed set of transformations as in our previous analysis depicted in Figure 1. We vary the intensity of the hue, the probability of color jitter, the size of the random crop, and the probability of horizontal inversion through 20 uniformly sampled values for each, and repeat each training five times with distinct seed values. We compute the Pearson, Kendall and Spearman correlation coefficients for each pair of classes with respect to a given transformation parameter, as well as their respective p-values, and define class pairs with opposite behaviors as those with at least one negative correlation score of the three measured correlations lower than -0.3 and a p-value lower than 0.05. We then measure the ratio of classes with at least

Table 1: **Correlation values between class properties and the effect of transformations on classes.** We focus on class properties such as Intrinsic Dimension, Texture Analysis, Fourier Transform, and Spectrum of Feature Covariance, and transformations such as Hue Intensity, Color Jitter Probability, Crop Size, applied on Cifar100. Values significantly larger than 1 indicate a notable difference between behavior groups with respect to the varying transformation. Asterisks (*) denote p-values > 0.05, indicating less significant correlations

| Class properties | Hue Intensity | Color Jitter Probability | Crop Size |
|---|---|---|---|
| Intrinsic Dimension | 16.64 | 19.15 | 0.21* |
| Texture Analysis | 16.39 | 4.89 | 0.18* |
| Fourrier Transform | 0.71* | 7 | 5.19 |
| Spectrum of Feature Covariance | 1.27 | 0.56* | 0.97* |

one opposite behavior to another class, compared to the total number of classes, in order to understand the extent of inter-class bias for a given transformation, method, and dataset.

Our findings, as represented in Figure 2a, indicate that the extent of inter-class bias for the self-supervised learning methods of interest varies among different transformations. This variability is primarily due to the fact that while these transformations aim to preserve the features that define a class across the original image and its transformed versions, they can also inadvertently compromise information specific to a particular class, while favoring the information of another class. Notably, within Cifar100, a dataset encompassing a diverse range of natural image classes, we observe a significant presence of inter-class bias when manipulating hue intensity. This outcome can be attributed to the optimization of specific features through each transformation choice, which may not be

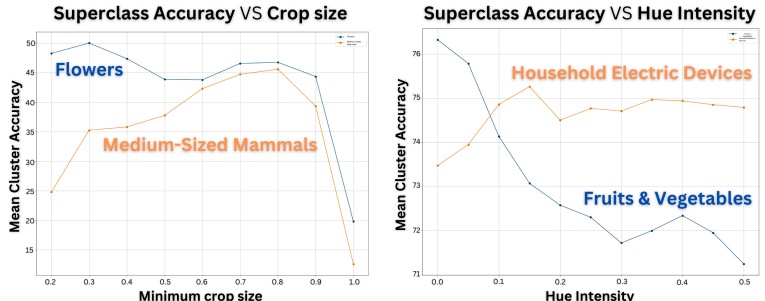

Figure 3: **Specific transformation choices control superclass performance.** A comparative analysis of the mean accuracy for superclasses in Cifar100, considering trainings conducted using BYOL, SimCLR, and VICReg as SSRL approaches, with variations in crop size or hue intensity. Noteworthy observations reveal consistent patterns across all models, illustrating how each transformation parameter can exert a distinct influence on different superclasses. Consequently, each superclass exhibits unique optimal parameters, emphasizing the potential for transformation parameter selection to effectively modulate the performance of specific superclasses.

optimal for certain classes. To substantiate the generality of these findings across convolution-based networks, we conduct a comparative analysis on Cifar100 using ResNet18, ResNet50, and ConvNext-Tiny as encoders, along with BYOL, SimCLR, and VICReg as the self-supervised learning approaches. By varying hue intensity, the results, as presented in Table 2b, reaffirm the consistent trend. An analysis on the amount of shared classes with negative correlations between the different SSRL approaches is done in Supplementary Materials Table 6.

To investigate the potential relationship between abstract class properties and their preferred transformations, we conduct a thorough analysis of each class's response to varying transformation parameters. We explore class accuracy behavior under distinct transformations, namely, Hue Intensity, Color Jitter Probability, and Crop Size. By computing the slope of the linear regression line that best fits the accuracy-transformation data for each class and each model, we categorize the behavior of each class accuracy as ascending, descending, or random. Simultaneously, we compute a texture analysis measure, and a Fourier transform measure for each class in the Cifar100 dataset, as well as the spectrum of feature covariance and the intrinsic dimension using the features resulting from a ResNet model pretrained on ImageNet in a supervised manner. Using Anova

Table 2: **Measured metrics for clustering, linear evaluation, and average LPIPS** Zhang et al. (2018) are presented for VGG11 models trained on the MNIST dataset LeCun et al. (1998) with MoCov2 Chen et al. (2020b) as SSRL approach, using distinct sets of transformations. The effects of specific transforms within each set are analyzed through different training configurations. In bold, the First Set results into a representations of digits, and the Second Set results into a representation focused on the handwriting style and line thickness. Top1 Accuracy is computed on a Linear Evaluation, and the LPIPS score is obtained using an AlexNet Krizhevsky et al. (2012) backbone. Lower scores indicate higher perceptual similarity between transformed image views. The Silhouette score Rousseeuw (1987) indicates relatively good cluster quality for the second set, despite the failure of its resulting clusters to accurately capture the digit clusters, as evident from the AMI score.

| TRANSFORMATION SETS | SILHOUETTE | AMI | TOP1 ACC | LPIPS |
|---|---|---|---|---|
| ROTATION+CROP | 0.74 | 0.79 | 98.4 | 0.22 |
| ROTATION+CROP+PADDING | 0.78 | 0.81 | 99.3 | 0.25 |
| **Rotation+Crop+Padding +ColorInversion (First Set)** | **0.87** | **0.83** | **99.6** | **0.33** |
| ROTATION+CROP+FLIPS | 0.71 | 0.66 | 96.2 | 0.32 |
| **Rotation+Crop+Flips+RandomErasing (Second Set)** | **0.66** | **0.37** | **62.1** | **0.51** |

and Manova correlation metrics, we then compute the correlation between these class properties and the class behaviors when varying a specific transformation.

Our results, summarized in Table 1, provide insights into the relationship between abstract image class properties and the effect of variations of transformation parameters. For instance, the Intrinsic Dimension and Texture Analysis of image classes exhibit substantial correlation with variation of Hue Intensity, implying that the intrinsic complexity and texture attributes of classes could significantly influence their response to changes in this transformation. A similar pattern is noticed with the Color Jitter Probability, albeit with a somewhat weaker correlation. Interestingly, the Spectrum of Feature Correlation shows minimal correlation with all transformations, suggesting that the covariance of class features might not significantly affect the class response to transformations. The Fourier Transform property showed mixed results, with a weak correlation with Hue Intensity but a stronger one with Crop Size, as the crop transformation can induce a varying degree of loss of signal in the image.

These results imply that the choice of transformations not only introduces a inter-class bias that can subtly impact performance in real-world scenarios, but it also presents an opportunity to harness this bias to achieve a desired balance in class performance or optimize specific class accuracies for specific use cases. We focus in the following analysis on the coarse grained labels of Cifar100, commonly called superclasses in the literature. As demonstrated in Figure 3, we observe that certain superclasses in the Cifar100 dataset exhibit improved recognition when specific transformation parameters are applied, when other don't. This highlights the potential of consciously selecting and studying transformations in our training process to enhance the performance of specific class clusters or achieve a balanced performance across classes. Therefore, the careful tailoring of specific transformations and their parameters becomes crucial in preserving desired information within classes, presenting a potential avenue for improvement in training.

### 3.2 Transformation choice impacts clustering and representation information

Examining the impact of transformations on diverse tasks, with variations in transformation parameters and compositions, is crucial for comprehensively understanding their influence on the quality of the resulting representations. In our study, we specifically concentrate on the unsupervised clustering task to elucidate how the choice of transformations affects the type of encoded information. To this end, we direct our investigation towards the representations that can be attained by training encoders with different architectures (VGG11, ResNet18, ConvNeXt-Tiny) with two prominent SSRL approaches (MoCov2 Chen et al. (2020b), BYOL Grill et al. (2020)) on the MNIST benchmark dataset LeCun et al. (1998). Our study delves into the effects of different sets of compositions of transformations on the nature of the information embedded within the representation and its correlation to the expected intent of our tasks. For our trainings, we employ two sets of transformations. The first set, comprised of padding, color inversion, slight rotation, and random

cropping, aims to maximize the intra-class overlap of digits. This is due to these transformations not affecting the digit information, and keeping it invariant in the transformed version of the images. The second set, which includes vertical flips, strong rotation, random cropping, and random erasing, enables us to investigate the representation resulting from the destruction of digit information in the transformed views, as these transformations destroy an integral part of the digit itself in the transformed version of the image. We similarly employ in additional trainings other transformation configurations sampled from each transformation set, to analyze their effects on the resulting representations. Each training iteration is repeated five times with distinct seeds, and the average of their scores is computed.

To facilitate a comprehensive comparison and evaluation of our findings, as well as to gauge the extent to which transformations can disrupt perceptual similarities between image views, we employ the Learned Perceptual Image Patch Similarity (LPIPS) metric Zhang et al. (2018). This metric allows us to measure the perceptual similarity between views after undergoing transformations. Following the training process, we conduct a K-Means clustering Lloyd (1982) with ten clusters, and a linear evaluation using the digit labels. In order to measure the efficacy of the clustering, we employ the Silhouette score Rousseeuw (1987). This metric calculates a measure of how close each sample in one cluster is to the samples in the neighboring clusters, and thus provides a way to assess data cluster quality. Higher Silhouette scores indicate that samples are well clustered and lower scores signify that samples are incorrectly clustered. Additionally, we use the Adjusted Mutual Information score (AMI) Vinh et al. (2010), a variation of Mutual Information that accounts for chance, providing a more robust evaluation of the clustering. The AMI score quantifies the agreement between the assigned cluster labels and the true labels, and is normalized against the expected Mutual Information to reduce its dependency on the number of clusters. A higher AMI score corresponds to a more accurate clustering with respect to the true labels. A more in-depth examination of the AMI score can be found in Supplementary materials, Section B.3.

As evident from the findings presented in Table 2, a progressive decline is observed in both the AMI score and the top1 accuracy score as we transition from the initial set of basic transformations (rotation and crop) to the second set of transformations. This decline is accompanied by a notable increase in the perceptual dissimilarity between transformed image views for the second set, which is to be expected considering the highly destructive nature of the random erasing transformation in comparison to random flips. Consequently, the resulting representation manifests a substantial reduction in the information associated with digits, as evidenced by the conspicuous decline in the accuracy of digit classification, which remains unmitigated despite the implementation of supervised training during linear evaluation. Nevertheless, it is worth noting that while the AMI score experiences a more pronounced decrease, the silhouette score exhibits a slight decline. This suggests that the clusters formed by the representations resulting from the second set of transformations remain well-separated and encode meaningful information beyond mere noise. As illustrated in Figure 4, the resulting clusters from both representations demonstrate distinguishable characteristics. In particular, the clusters derived from the second set of transformations capture handwriting attributes such as

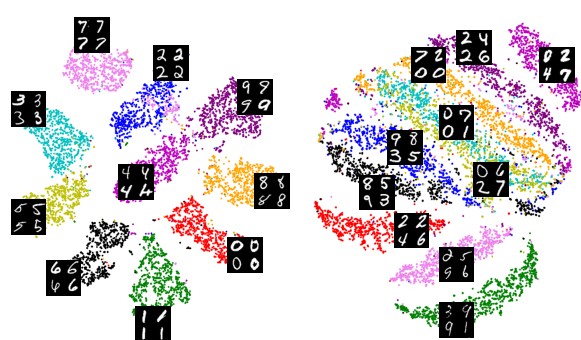

Figure 4: **The selection of transformations dictates the features learnt during the training process, thus enabling the adaptation of a model for different tasks.** A t-SNE projection of the ten-class clustering of the MNIST dataset LeCun et al. (1998) performed on two representations obtained from two self-supervised trainings of the same model using MoCo V2 Chen et al. (2020b), with the sole distinction being the selection of transformations employed. One representation retains information pertaining to the digit classes *(left)*, achieved through padding, color inversion, rotation, and random cropping, while the other representation preserves information regarding the handwriting classes *(right)*, achieved through vertical flips, rotation, random cropping.

line thickness and writing flow, effectively forming distinct handwriting classes where the transformations maximize intra-class variance while minimizing inter-class overlap. Additional results for alternative backbone architectures and SSRL approaches are provided in Table 8 and Figure 12 in the Supplementary Materials,

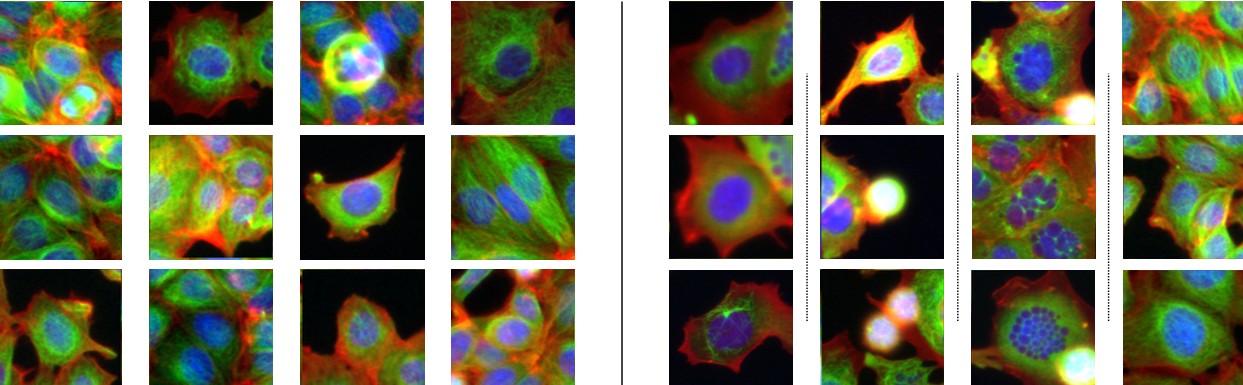

Figure 5: **As a result of their unique genetic expression and surrounding environment, single cells** *(left - untreated cells)* **are already substantially dissimilar within a given condition, complicating the identification and measurement of various perturbations.** The figure illustrates *(right - cells treated with high concentration of Nocodazole)* that a given treatment can produce four morphological responses (each column), with one of which can appear similar to untreated cells (most right). The majority of treatments at lower concentrations yield phenotypes that are indistinguishable by the naked eye from untreated cells (data not shown). The images of this dataset are cropped such that there is always a cell at the center.

and consistently demonstrate similar trends throughout the conducted experiments. These observations indicate that, beyond enhancing the performance of specific classes, we can selectively supervise and encode desired image features into our representations by conscientiously selecting the appropriate transformations during training. This deliberate or unconscious choice of transformations serves to improve the overall performance of the trained model for the given task.

### 3.3 The effect of transformations correlates with the subtlety of a domain

With the objective of exploring the amplitude of the effect of transformations on domains characterized by inherently subtle dissimilarities among class images, we conduct experiments on microscopy images of cells under two conditions (untreated vs treated with a compound). These conditions are available from BBBC021v1 Caie et al. (2010), a dataset from the Broad Bioimage Benchmark Collection Ljosa et al. (2012). The dataset consists of cells that demonstrate heterogeneous variability in their appearance, even when in the same condition. Notably, both the within-condition variability and the visual disparities between conditions exhibit subtle characteristics. This context poses a more demanding challenge for self-supervised representation learning, as illustrated in Figure 5. In order to observe the effect of transformations in such a context, we preprocess these microscopy images by detecting all cell nuclei and extracting an 196x196 pixels image around each of them. We focus our study on three main compounds : Nocodazole, Cytochalasin B and Taxol. Technical details of the dataset used and the data preprocessing performed can be found in Supplementary Material A. We use a VGG13 Simonyan & Zisserman (2015) and a ResNet18 encoder architecture, with MoCov2 Chen et al. (2020b), BYOL Grill et al. (2020) and VICReg Bardes et al. (2022a) as the self supervised approaches, and run two separate trainings of the model from scratch for each compound, each of the two trainings with a different composition of transformations for invariance, repeated five times with distinct seeds. We then perform a K-Means Lloyd (1982) clustering (k=2) on the inferred test set embeddings and compute the Adjusted Mutual Information score (AMI) Vinh et al. (2010) with respect to the ground truth compound labels of the compounds data subsets (untreated vs treated with Nocodazole, untreated vs treated with Cytochalasin B, untreated vs treated with Taxol).

For each composition of transformations explored, we repeat the training five times with different seeds, and compute the average and standard deviation of the AMI score (technical details of the the model training can be found in Supplementary material Section B.4). Table 3 displays the AMI scores achieved by using two different compositions of transformations in training, compared to the AMI score of a clustering on

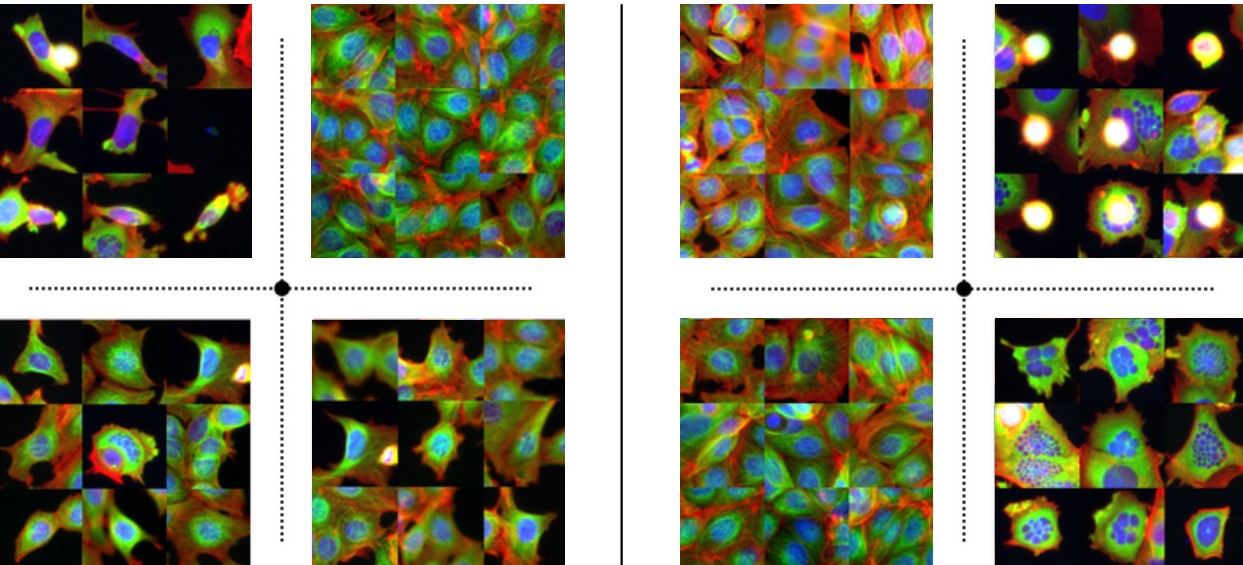

Figure 6: **An illustration of the various K-Means (k=4) clustering results on the data subset of Nocodazole** defined in Supplementary Material A, utilizing different combinations of augmentations with a VGG13 as backbone and MoCov2 as SSRL approach, the objective being to separate the distinct morphological reactions of the cells into different clusters. The sampled images are the closest images to the centroid of the cluster using Euclidean distance in the representations. The clusters on the left predominantly take into consideration the number of cells present in each image, and result from training the model with a combination of color jitter, flips, rotation, affine transformation, and random cropping. The clusters on the right take into account some of the phenotypes under examination, and result from training the model with a combination of rotations, center cropping, color jitter, and flips. The parameters of each transformation are detailed in Supplementary Material B.4.

representations achieved with a pretrained model trained on ImageNet with supervision. By replacing slight random cropping and resizing, a transformation used in most existing self supervised approaches, with very strong random rotations of the image, we report a significantly higher mean AMI score, which shows that the model using random rotations is able to learn representations that better separate untreated from compound treated cells, including cells displaying subtle differences unnoticeable by the naked eye. Inversely, random cropping was destructive to the sought out information in this case, as the cropped images can miss out on relevant features in the sides of the cell. In contrast to the effects of transformations reported on the overall accuracy of datasets with less subtle class differences, as discussed in Section 3.1, the markedly greater impact observed on this specific type of data implies that transformations can exert a more substantial influence on the learning of more effective representations, which capture the full range of image variability within datasets characterized by subtle distinctions between classes. By optimizing the selection of transformations for specialized goals on such datasets, competitive performance comparable to models pretrained with supervision can be achieved, even in the context of a relatively small-scale dataset.

Beyond clustering into two conditions, we wonder what combination of transformations could lead to a proper clustering of cell phenotypes (or morphology). We explore different compositions of transformations in additional experiments with the same VGG13 Simonyan & Zisserman (2015) architecture and MoCov2 Chen et al. (2020b) loss function. We then apply K-Means Lloyd (1982) clustering (k=4) on the representations obtained from the test set. As observed in Section 3.2, different compositions of transformations can lead to very different clustering results. This is further confirmed in the microscopy domain and well illustrated by observing the Nocodazole treatment (Figure 6) where the composition of color jitter, flips, rotation, affine, in addition to random crops, results in clustering images by the number and size of cells, rather than by morphological features (*Figure 6 left*). We perform a training where affine transform and random crop were replaced by a center crop that preserves 50% of the image around the central cell. The latter resulted in four clusters where two out of the three cell phenotypes were detected. However, it also had the effect of splitting untreated cells into two different clusters (*Figure 6 right*). This aligns with the findings

Table 3: **The results of the adjusted mutual information score Vinh et al. (2010) obtained for two sets of transformations, with different SSRL approaches and backbones**, through the mean of five training runs for each, compared to each other and to the AMI score achieved on the representations of pre-trained models (Resnet 101 and VGG16) trained with supervision on ImageNet, and applied on the dataset subsets containing Nocodazole, Cytochalasin B and Taxol. The selection of the pretrained models width is studied in Supplementary Materials B.4. Both sets of transformations comprise random rotations, affine transformations, color jitter and flips, with the first set including an additional random cropping, and resulting in a mediocre AMI score, and the second set applying random rotations and resulting in a significantly higher score.

| Transformations | SSRL approach | Backbone | Nocodazole | Cytochalasin B | Taxol |
|---|---|---|---|---|---|
| First Set : | MoCo v2 | VGG13 | 0.19 | 0.27 | 0.16 |
| | | ResNet18 | 0.17 | 0.25 | 0.15 |
| | Byol | VGG13 | 0.21 | 0.28 | 0.19 |
| | | ResNet18 | 0.2 | 0.25 | 0.17 |
| | VICReg | VGG13 | 0.19 | 0.26 | 0.2 |
| | | ResNet18 | 0.16 | 0.25 | 0.21 |
| Second Set : | MoCo v2 | VGG13 | 0.37 | 0.45 | 0.38 |
| | | ResNet18 | 0.33 | 0.42 | 0.31 |
| | Byol | VGG13 | 0.38 | 0.48 | 0.41 |
| | | ResNet18 | 0.35 | 0.44 | 0.34 |
| | VICReg | VGG13 | 0.38 | 0.44 | 0.36 |
| | | ResNet18 | 0.34 | 0.43 | 0.3 |
| Pretrained models on ImageNet | | VGG16 | 0.34 | 0.55 | 0.36 |
| | | ResNet101 | **0.39** | **0.57** | **0.43** |

presented in Section 3.2, as the diverse clustering outcomes mirror the distinct transformation approaches taken to encode the intrinsic information embedded within the images of this dataset. The presence of subtle variations within this dataset underscores the heightened sensitivity to the selection of transformations, which amplifies the multitude of potential representations accordingly. Altogether, engineering a combination of transformations in this context represents a somewhat weak supervision that can become a silent but strong bias or, alternatively, can be leveraged as a powerful tool to achieve a desirable result on a specialized task.

### 3.4 A delineation of desired features enhances task-specific representations.

Upon visual inspection of the cell distributions depicted in Figure 5 pertaining to both untreated and Nocodazole-treated cells, we postulate that the accurate segregation of distinct subtle cellular responses, and subsequently comprehending the influence of different compounds on cellular morphology, hinges upon the alterations occurring within the cell, including the nucleus, as well as their influence on intercellular interactions. Guided by this hypothesis, we explicitly delineate the specific features of interest for extraction from the images. Our primary focus lies on capturing the diverse morphological characteristics exhibited by various components of the cell, while also considering the spatial relationships between the central cell and its surrounding counterparts, albeit to a slightly lesser extent. Consequently, we validate the aforementioned hypothesis through distinct training experiments conducted on three compound data subsets: Nocodazole, Taxol, and Cytochalasin B.

By adhering to our rigorous definition of the desired features, we endeavor to establish transformation compositions that preserve these features in the transformed views. In addition to incorporating Affine transformations and color jitter and random rotations, we employ random center cropping to focus on the image center (which aligns with the center of a cell given the preprocessing step). This approach enables the model to learn morphology-related features specific to the cells themselves. To emphasize intercellular interactions, we devise using random cropping on the images. However, certain incompatible combinations of transformations emerge, such as using random cropping with center cropping or vice versa, thereby undermining the preservation of crucial information.

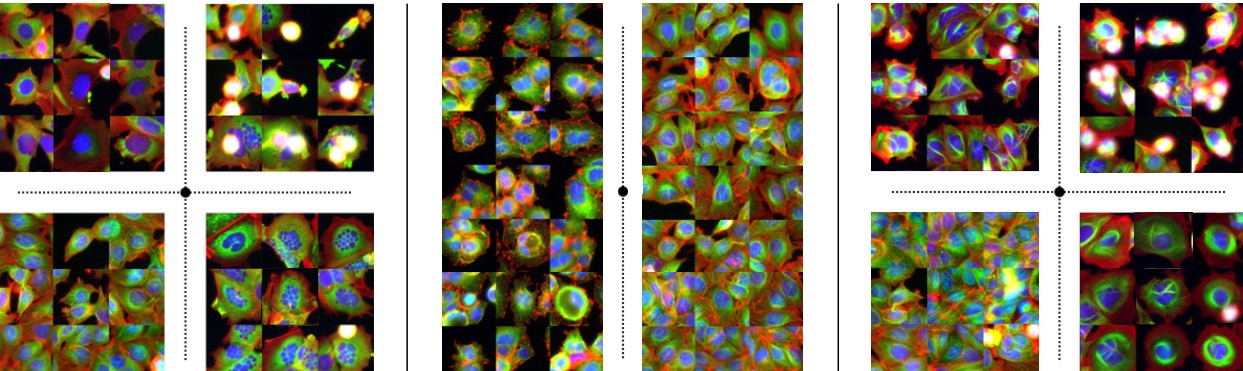

Figure 7: **The clustering results achieved through the utilization of two MoCo v2 losses Chen et al. (2020b) with a VGG13 backbone, each with a distinct set of transformations**, on the Nocodazole *(left)*, Cytochalasin B *(middle)*, and Taxol *(right)* image treatment subsets. One loss employs color jitter, flips, rotation, affine transformation, and random cropping, while the other uses rotations, center cropping, color jitter, and flips. The clustering results demonstrate that the phenotypes of each subset are clearly separated and represented in each cluster, as evidenced by the images closest to its centroïd.

Consequently, we define two distinct transformation compositions, each corresponding to the compositions depicted in Figure 6, and keep both separated in two independent SSRL losses of the same SSRL approach. Subsequently, we train a model to minimize the weighted sum of these two losses on each of the compound data subsets. Following training, we perform K-Means clustering (k=4) on the resulting representations for the Nocodazole and Taxol subsets, and K-Means clustering (k=2) on the resulting representation for the Cytochalasin B subset. Further details on the parameters of the weighted sum are available in Supplementary Materials B.4.

In the subsequent results in Figure 7 we can observe that we successfully separate all cell pheno-types/morphological alterations obtained after each of the three compound treatments, from each other and from untreated cells, which validates qualitatively our hypothesis of the features of interest. By performing another K-Means (k=2) on the same representations, we also report AMI scores to analyze the separation level of the compound-treated cells from untreated cells in the representation space. The resulting scores in Table 4 can be considered quite high in this context where treated cells can look like untreated cells, vastly

Table 4: **A comparison of the AMI score of the clusterings** achieved through a K-Means (k=2) over the representations of models trained through our combination of two SSRL losses and the representations of an ImageNet pre-trained encoder, on the compound treated cell image subsets Nocodazole, Cytochalasin B, and Taxol. One loss employs color jitter, flips, rotation, affine transformation, and random cropping, while the other utilizes rotations, center cropping, color jitter, and flips. Through cautious selection of a combination of different sets of transformations that are adapted to the features we seek to extract, we surpass on clustering the performance of models pretrained with supervision, using a self supervised training, even with small scale datasets.

| Transformations | SSRL approach | Backbone | Nocodazole | Cytochalasin B | Taxol |
|---|---|---|---|---|---|
| Weighted Combination of Sets | MoCo v2 | VGG13 | 0.51 | 0.66 | 0.52 |
| | | ResNet18 | 0.46 | 0.63 | 0.47 |
| | Byol | VGG13 | 0.51 | 0.64 | **0.54** |
| | | ResNet18 | 0.47 | 0.61 | 0.48 |
| | VICReg | VGG13 | **0.55** | **0.67** | 0.51 |
| | | ResNet18 | 0.5 | 0.63 | 0.45 |
| Pretrained models on ImageNet | | VGG16 | 0.34 | 0.55 | 0.36 |
| | | ResNet101 | 0.39 | 0.57 | 0.43 |

surpassing previous trainings with separate compositions of transformations in Table 3, as well as surpassing the performance of pretrained models even with a small scale dataset.

For a more comprehensive analysis of the results in Table 4, we conduct a more systematic examination of the interplay and impact of each of the two compositions of transformations on the combined performance of the weighted sum of losses, by modifying each combination before incorporating it with the other transformation during the Nocodazole training. The results presented in Figure 8 reveal substantial variations in the scores and, consequently, the resulting feature representations among different parts of the transformation combinations. Particularly noteworthy is the enhancement in performance when gradually incorporating transformations into the second set, which focuses on rotation-invariant features and the cellular center, while keeping the first set fixed. In contrast, the reverse process of gradually adding transformations to the first set, while keeping the second set constant, does not yield iterative improvements. This observation suggests that our defined features for this task accurately capture the significance of the cellular center, which remains invariant under rotation transformations or center cropping, in effectively separating treated cells from untreated ones. Furthermore, it demonstrates that the surrounding cells also play a role, albeit to a lesser extent, in achieving optimal representation for the task. These findings underscore the potential of more intricate transformation manipulations, beyond single parameter modifications, to yield superior rep-

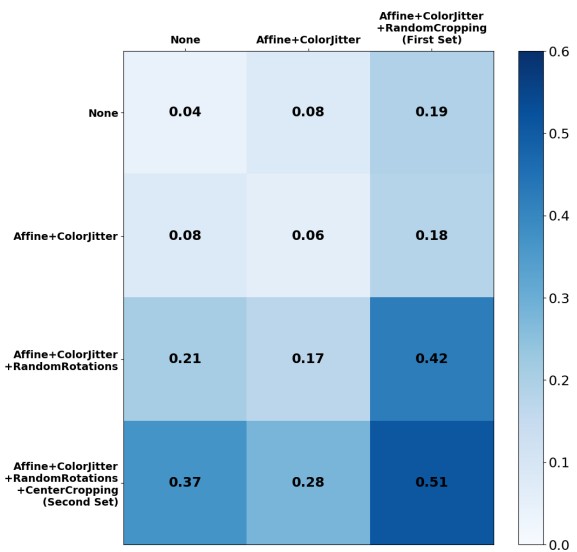

Figure 8: **An ablation study examining the Adjusted Mutual Information (AMI) score results** obtained by progressively incorporating different transformations, with MoCo v2 SSRL approach and a VGG13 backbone. Notably, the combination of the first set of transformations with random rotations significantly enhances the score, and the resulting representation accordingly. Moreover, the inclusion of center cropping, which emphasizes the cellular core, yields the most favorable outcomes.

resentations that significantly enhance task performance. However, identifying the optimal combination necessitates a rigorous definition of the sought-after features, which, although label-free, provides a form of weak supervision.

## 4   Conclusion

In this work, we delve into the impact of transformation choices on convolution-based approaches, specifically focusing on small to medium-scale datasets. Our experiments demonstrate that the selection, magnitude, and combination of transformations significantly influence the efficacy of self-supervised representations. Notably, the choice of transformations not only introduces inter-class bias but also offers a powerful tool for controlling and balancing class performances. Moreover, by carefully selecting and defining transformations to optimize the encoding of specific features, we achieve improvements in task-specific performance. Importantly, our findings emphasize the amplified consequences of transformation choices in domains characterized by fuzzy or less visually distinguishable class differences, surpassing the performance of pretrained models on small-scale datasets. Altogether, some of the results can be understood somewhat intuitively. If one erases color from a car dataset, a deep network might not find enough correlated information to be able to classify cars on the basis of their original color. Thus the question: what is a good representation? In scenarios where massive datasets are not readily available, the correct answer is that it depends on the desired task. Although the initial goal of SSRL was to circumvent such circumstances, our findings in this particular context reveal the efficacy that can be achieved by judiciously selecting an appropriate combination of transformations, informed by a profound comprehension of the most salient features.

This study acknowledges several limitations, such as the utilization of small to medium-scale datasets and convolution-based approaches. Future research should consider examining larger datasets and exploring the potential benefits of combining them with transformers. Furthermore, expanding the analysis to include downstream task performance could provide a more comprehensive understanding of the impact of transformations. A promising perspective lies in the potential use of informed transformation choices to fine-tune foundational models for specific tasks, even in the absence of labeled data. Overall, this study contributes valuable insights and suggests promising avenues for future investigations in the field of transformation learning within deep learning frameworks.

Our research underscores the importance of thoughtful transformation selection in self-supervised learning, encouraging a more discerning approach to hyperparameter choice. This may inspire a shift from automatic or blind selection towards a more principled understanding of augmentations, potentially leading to more robust and nuanced model performances across diverse domains. We see no significant negative ethical implications at this time.

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

# A    Datasets

We perform the experiments mentioned in Section 3.3 on microscopy images available from BBBC021v1 Caie et al. (2010), a dataset from the Broad Bioimage Benchmark Collection Ljosa et al. (2012). This dataset is composed of breast cancer cells treated for 24 hours with 113 small molecules at eight concentrations, with the top concentration being different for many of the compounds through a selection from the literature. Throughout the quality control process, images containing artifacts or with out of focus cells were deleted, and the final dataset totalled into 13,200 fields of view, imaged in three channels, each field composed of thousands of cells. In the cells making up the dataset, twelve different primary morphological reactions from the compound-concentrations were identified, with only six identified visually, while the remainder were defined based on the literature, as the differences between some morphological reactions were very subtle. We perform a simple cell detection in each field of view, in order to crop cells centered in (196x196px) images. We then filter the images by compound-concentration, and keep images treated by Nocodazole at its 4 highest concentrations. These compound-concentrations result in 4 morphological cell reactions, for which we don't have individual labels for each cell image. We sample the same number of images from views of untreated cells, totalling into a final data subset of 3500 images, which we split into 70% training data, 10% validation data and 20% test data. We repeat the same process for the compounds Taxol and Cytochalasin B, each containing 4 and 2 morphological reactions respectively, to result in data subsets of sizes 1900 and 2300 images, respectively.

# B    Model training

## B.1    Study of inter-class bias in self supervised classification

For results in Section 3.1 and Table 5, we run several trainings of 12 SSRL methods  Bardes et al. (2022a); Caron et al. (2018; 2020); Chen et al. (2020a); Chen & He (2021); Chen et al. (2020b); Dwibedi et al. (2021); Grill et al. (2020); Lee & Aune (2021); Zbontar et al. (2021); Zheng et al. (2021) on Cifar10 and Cifar100 Krizhevsky (2009) with a Resnet18 architecture, with a pretraining of 1000 epochs without labels. We use the same transformations with similar parameters on all approaches, namely a 0.4 maximal brightness intensity, 0.4 maximum contrast intensity, 0.2 maximum saturation intensity, all with a fixed probability of 80%. With a maximal hue intensity of 0.5, we vary the hue probability of application between 0% and 100% by uniformly sampling 20 probability values in this range. We use stochastic gradient descent as optimization strategy for all approaches, and through a line search hyperparameter optimization, we use a batch size of 512 for all approaches except Dino Caron et al. (2021) and Vicreg Bardes et al. (2022a) for which we use a batch size of 256. Following the hyperparameters used in the literature of each approach, we use a projector with a 256 output dimension for most methods, except for Barlow Twins Zbontar et al. (2021), Simsiam Chen & He (2021), Vicreg Bardes et al. (2022a) and Vibcreg Lee & Aune (2021), for which we use a projector with a 2048 output dimension, and DeepclusterV2 Caron et al. (2018) and Swav Caron et al. (2020) for which we use a projector with a 128 output dimension. For some momentum based methods (Byol Grill et al. (2020), MocoV2+ Chen et al. (2020b), NNbyol Dwibedi et al. (2021) and Ressl Zheng et al. (2021)), we use a base Tau momentum of 0.99 and use a base Tau momentum of 0.9995 for Dino Caron et al. (2021). For Mocov2+ Chen et al. (2020b), NNCLR Dwibedi et al. (2021) and SimCLR Chen et al. (2020a), we use a temperature of 0.2.

We train the Barlow Twins Zbontar et al. (2021) based model with a learning rate of 0.3 and a weight decay of $10^{-4}$, and Byol Grill et al. (2020) as well as NNByol Dwibedi et al. (2021) with a learning rate of 1.0 and a weight decay of $10^{-5}$. We train DeepclusterV2 Caron et al. (2018) with a learning rate of 0.6, 11 warmup epochs, a weight decay of $10^{-6}$, and 3000 prototypes. We train Dino with a learning rate of 0.3, a weight decay of $10^{-4}$, and 4096 prototypes, while we train MocoV2+ Chen et al. (2020b) with a learning rate of 0.3, a weight decay of $10^{-4}$ and a queue size of 32768. For NNclr Dwibedi et al. (2021), we use a learning rate of 0.4, a weight decay of $10^{-5}$, and a queue size of 65536. We train ReSSL Zheng et al. (2021) with a learning rate of 0.05, and a weight decay of $10^{-4}$, while we train SimCLR Chen et al. (2020a) with a learning rate of 0.4, and a weight decay of $10^{-5}$. For Simsiam Caron et al. (2020), we use a learning rate of 0.5, and a weight decay of $10^{-5}$, and use for Swav Caron et al. (2020) a learning rate of 0.6, a weight decay of $10^{-6}$, a

queue size of 3840, and 3000 prototypes. We use for Vicreg Bardes et al. (2022a) and Vibcreg Lee & Aune (2021) a learning rate of 0.3, a weight decay of $10^{-4}$, an invariance loss coefficient of 25, and a variance loss coefficient of 25. We use a covariance loss coefficient of 1.0 for Vicreg Bardes et al. (2022a) and a covariance loss coefficient of 200 for Vibcreg Lee & Aune (2021). We perform linear evaluation after each pretraining for all methods, through freezing the weights of the encoder and training a classifier for 100 epochs. We use 5 different global seeds (5, 6, 7, 8, 9) for each hue intensity value, and compute the mean top1 accuracy resulting from the linear evaluation, using each of the 5 different experiences. Each training run was made on a single V100 GPU. On ImageNet100 Deng et al. (2009), we train a Resnet18 encoder with BYOL Grill et al. (2020), MoCo V2 Chen et al. (2020b), VICReg Bardes et al. (2022a) and SimCLR Chen et al. (2020a), using a batch size of 128 for 400 epochs. We use a learning rate of 0.3 and a weight decay of $10^{-4}$ for MoCo V2 and VICReg, and a weight decay of $10^{-5}$ and learning rates of 0.4 and 0.45 for SimCLR and BYOL respectively. We repeat each experience three times with three global seeds (5,6 and 7) and compute its mean and standard deviation. We repeat the same experiment with the same parameters on ResNet50 and ConvNeXt-Tiny.

For the results in Figures 1 and 2a, we reuse the same training hyperparameters for Barlow Twins Zbontar et al. (2021), Moco V2 Chen et al. (2020b), BYOL Grill et al. (2020), SimCLR Chen et al. (2020a) and Vicreg Bardes et al. (2022a), and uniformly sample 10 values in the range of [0;0.5] for the maximal hue intensity, with a fixed 80% probability. We run different experiments for the 5 global seeds for each hue intensity, and compute their mean and standard deviation. We perform the same process while fixing maximal hue intensity to 0.1, and varying its probability by uniformly sampling 20 probability values in the range [0;100]. We repeat a similar process for the random cropping and horizontal flips, by sampling 8 values uniformly in the range of [20;100] of the size ratio to keep of the image, and sampling 20 values uniformly in the range of [0;100] for the probability of application of horizontal flips.

## B.2 MNIST Clustering

For the displayed clustering results in Figure 4, we use a VGG11 Simonyan & Zisserman (2015) architecture, with a projector of 128 output dimension, trained using a MocoV2+ Chen et al. (2020b) loss function on Mnist LeCun et al. (1998) for 250 epochs. We use an Adam optimizer, a queue size of 1024, and a batch size of 32. We set temperature at 0.07, learning rate at 0.001, and weight decay at 0.0001. We run two trainings with two separate sets of compositions of transformations, each run on a single V100 GPU, and perform a Kmeans (K=10) clustering on the resulting representations of the test set. For the digit clustering, we use a composition of transformations composed of a padding of 10% to 40% of the image size, color inversion, rotation with a maximal angle of 25°, and random crop with a scale in the range of [0.5;0.9] of the image, and then a resizing of the image to 32x32 pixels. For the handwriting flow clustering, we use a composition of transformations composed of horizontal & vertical flips with an application probability of 50% each, rotations with a maximal angle of 180°, random crop with a scale in the range of [0.9;1.1], and random erasing of patches of the image, with a scale in the range of [0.02;0.3] of the image and a probability of 50%. We perform linear evaluation by training classifiers on the frozen representations of the trained models, in order to predict the digit class, and evaluate using the top1 accuracy score. For results on Table 8 and Figure 12, we use ResNet18 and ConvNeXt-Tiny architectures with BYOL and MoCov2 ass SSRL approaches. We use for BYOL a learning rate of 0.01 and a weight decay of $10^{-5}$, with a projector with a 256 output dimension. We then use the same parameters and augmentations as previous trainings.

## B.3 Clustering evaluation with the AMI Score

We use the adjusted mutual information (AMI) Vinh et al. (2010) in Sections 3.3 and 3.4 to evaluate clustering quality, and to measure the similarity between two clusterings. It is a value that ranges from 0 to 1, where a higher value indicates a higher degree of similarity between the two clusterings. This score holds an advantage over clustering accuracy Kuhn (1955) in one main aspect, being that the clustering accuracy only measures how well the clusters match the labels of the true clusters, and does not take into account the structure within the clusters, such as heterogeneity of some of the clusters. This is unlike the AMI score, which takes into account both the structure between the clusters and the structure within the clusters, by measuring the "agreement" between the groupings of a predicted cluster and the groupings of the true cluster. If both clusterings agree on most of the groupings, then the AMI score will be high, and inversely low if they do not.

The AMI score can be computed with the formula :

$$AMI(X,Y) = \frac{MI(X,Y) - E(MI(X,Y))}{max(H(X), H(Y)) - E(MI(X,Y))}$$

Where $MI(X,Y)$ is the mutual information between the two clusterings, $E(MI(X,Y))$ is the expected mutual information between the two clusterings, $H(X)$ is the entropy of the clustering $X$, and $H(Y)$ is the entropy of the clustering $Y$. Mutual information (MI) is a measure of the amount of information that one variable contains about another variable. In the context of AMI, the two variables are the clusterings $X$ and $Y$. $MI(X,Y)$ is a measure of to what extent the two clusterings are related to each other. Entropy is a measure of the amount of uncertainty in a random variable. In the context of AMI, the entropy of a clustering ($H(X)$ or $H(Y)$) is a measure of how much uncertainty exists within the clustering. Expected mutual information ($E(MI(X,Y))$) is the average mutual information between the two clusterings, assuming that the two clusterings are independent.

The adjusted mutual information (AMI) is calculated by first subtracting the expected mutual information ($E(MI(X,Y))$) from the actual mutual information ($MI(X,Y)$). This results in a measure of of the extent of the relationship between the two clusterings beyond what would be expected by chance. This value is then divided by the difference between the maximum possible entropy ($max(H(X), H(Y))$) and the expected mutual information ($E(MI(X,Y))$). Normalization of the result is achieved through this process, ensuring that it is always between 0 and 1. Figure 9 shows the results of a clustering achieved on Nocodazole vs untreated cells, with the AMI score computed after randomisation of the ground truth labels, in contrast to clustering results achieved without randomizing the labels.

### B.4 Cellular Clustering

For the results in Sections 3.3, 3.4, we use a VGG13 Simonyan & Zisserman (2015) architecture, trained using a MocoV2+ Chen et al. (2020b) loss function on the data subsets of the microscopy images available from BBBC021v1 Caie et al. (2010), presented in Section A, with a batch size of 128, for 400 epoch. We use an Adam optimizer, a queue size of 1024, and set temperature at 0.07, learning rate at 0.001, and weight decay at 0.0001. Each training is made on a single V100 GPU. We perform a Kmeans (K=2) on the resulting representations of the test set of the data subsets of Nocodazole, Cytochalasin B and Taxol, and evaluate the quality of the achieved clusters compared to the ground truth using the AMI score Vinh et al. (2010).

In Table 3, we achieve the first AMI result through usage of an affine transformation composed of a rotation with an angle of 20°, a translation of 0.1 and a shear with a 10° angle, coupled with color jitter with a brightness, contrast and saturation intensity of 0.4, and a hue intensity of 0.125, with a 100% probability, as well as random cropping of the image with a scale in the range of [0.9;1.1] and resizing to original image size of 196x196 pixel. For the second row result, we use an affine transformation composed of a rotation with an angle of 20°, a translation of 0.1 and a shear with a 10° angle, coupled with color jitter with a brightness, contrast and saturation intensity of 0.4, and a hue intensity of 0.125, with a 100% probability, and a random rotation with a maximal angle of 360°. The results achieved through a Resnet101, are achieved by performing a Kmeans (K=2) on the representations achieved on the compound subsets using the Resnet101, and evaluating the cluster assignment quality compared to ground truth using the AMI score Vinh et al. (2010). The choice of Resnet101 over other Resnet sizes is motivated through testing the performance of different sizes of Resnets on the different concentrations of Nocadozole/untreated cells, on which Resnet101 consistently shows the highest performance on the 4 highest concentrations, as shown in Figure 9.

For the clusterings in Figure 6, we train the same architecture with the same hyperparameters on different compositions of transformations, and perform a Kmeans (K=4) on the resulting representations of the test set. For all the clusters, the images displayed are the images closest to the centroïd of each cluster using an euclidean distance. The clustering in Figure 6 *left* is achieved by using a composition of color jitter with a brightness, contrast and saturation intensity of 0.4, and a hue intensity of 0.125, with a 100% probability, and horizontal and vertical flips, each with 50% probability of application, as well as random rotations with a maximal angle of 360°, an affine transformation composed of a rotation with an angle of 20°, a translation of

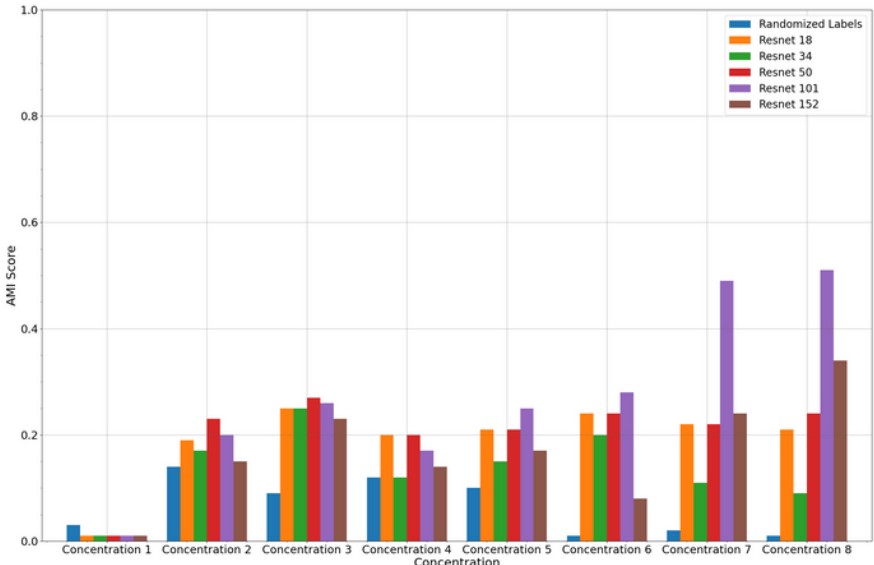

Figure 9: We perform a Kmeans clustering (K=2) on the Nocodazole and untreated images using Resnet models of various sizes, and evaluate them using the AMI score Vinh et al. (2010) on a similar number of randomly sampled images of each concentration of Nocodazole and a similar number of untreated cell images. We observe that Resnet101 outperforms the other Resnet model sizes. We also perform an experiment to better interpret the AMI scores achieved, by randomizing the labels and performing a clustering with Resnet101, and reporting the AMI score of the clustering compared to the randomized labels.

0.1 and a shear with a 10° angle, and a random crop with a scale sampled in the range [0.9;1.1], followed by a resizing of the image to the original size. The clustering in Figure 6 *right* is achieved by color jitter with a brightness, contrast and saturation intensity of 0.4, and a hue intensity of 0.125, with a 100% probability, horizontal and vertical flips, each with 50% probability of application, random rotations with a maximal angle of 360°, and a center crop with a scale of 0.5, followed by a resizing of the image to the original size. For the clustering in Figure 7, we trained the model with a sum of the losses (and corresponding transformations) of the models used in Figure 6. Through a gridsearch hyperparameter optimization, with the goal of optimizing the AMI score of a Kmeans clustering (K=2), we attributed a coefficient of 0.4 to the loss of the model used in Figure 6 *left*, and a coefficient of 1.0 to the loss of the model used in Figure 6 *right*.

## C   Additional results

Table 5: The mean and standard deviation of the top1 linear evaluation accuracy, obtained through the training of a Resnet18 architecture using 12 self-supervised approaches on the Cifar10 dataset, are presented. The approaches include VicReg Bardes et al. (2022a), DeepCluster v2 Caron et al. (2018), SWAV Caron et al. (2020), SimCLR Chen et al. (2020a), SimSiam Chen & He (2021), MoCo Chen et al. (2020b), NNCLR Dwibedi et al. (2021), BYOL Grill et al. (2020), VIBCReg Lee & Aune (2021), Barlow Twins Zbontar et al. (2021), and ResSL Zheng et al. (2021). The experiment involves the uniform sampling of 20 values for the hue transformation probability, while maintaining a fixed maximal intensity of 0.5, and all other transformation parameters are kept constant. The results indicate that despite the variation in the transformation probability, the overall accuracy of each method remains relatively consistent, with a minimal standard deviation value.

|  | Barlow Twins | Byol | Deep Cluster v2 | MoCo V2+ | nnByol | nnclr | Ressl | SimCLR | SimSiam | SwaV | Vibcreg | Vicreg |
|---|---|---|---|---|---|---|---|---|---|---|---|---|
| Mean | 89.59 | 92.09 | 86.9 | **92.37** | 91.3 | 89.76 | 90.21 | 90.16 | 89.6 | 86.96 | 82.47 | 89.82 |
| Std | 0.73 | **0.37** | 1.9 | 0.44 | 0.57 | 0.74 | 0.85 | 0.87 | 1.01 | 1.2 | 0.89 | 0.94 |

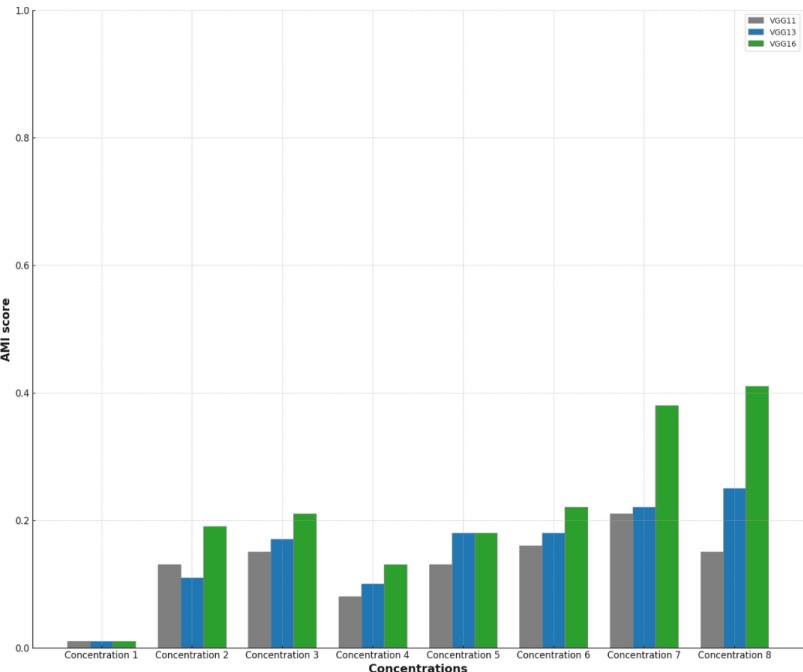

Figure 10: We perform a Kmeans clustering (K=2) on the Nocodazole and untreated images using VGG models of various sizes, and evaluate them using the AMI score Vinh et al. (2010) on a similar number of randomly sampled images of each concentration of Nocodazole and a similar number of untreated cell images. We observe that VGG16 outperforms the other VGG model sizes.

Table 6: Shared Classes with Inter-class Bias Across Different Self-Supervised Learning (SSL) Approaches. This table showcases the number of shared classes demonstrating inter-class bias within varying SSL methods, considering three key transformation parameters: Hue Intensity, Color Jitter Probability, and Crop Size. The SSL methods included in this analysis are Barlow Twins, BYOL, MoCo v2, SimCLR, and VICReg, all trained using a ResNet18 on Cifar100. The results highlight the degree of shared inter-class bias and the influence of different transformations, reinforcing the notion that class-level biases can be transformation and SSL-method dependent.

| Number of shared classes with inter-class bias | Hue Intensity | Color Jitter Probability | Crop size |
|---|---|---|---|
| In all 5 SSL approaches | 51 | 0 | 0 |
| In a minimum of 3 SSL approaches | 97 | 3 | 4 |
| In a minimum of 2 SSL approaches | 99 | 27 | 8 |

Table 7: Comparison of classes with significant negative correlations under variations of Hue Intensity for Linear Evaluation and Fine-tuning phases. The table displays the number of classes with statistically significant negative correlations (p-value < 0.05) for both Linear Evaluation and Fine-tuning under different SSL methodologies, SimCLR, BYOL, and VicReg, all with ResNet18 as the backbone. The last row represents the percentage of shared classes between Linear Evaluation and Fine-tuning that exhibited the same behavior trend (ascending, descending, or random).

| Methodology | Simclr | BYOL | VicReg |
|---|---|---|---|
| Resnet18 + Linear Evaluation | 97 | 80 | 92 |
| Resnet18 + Finetuning | 96 | 100 | 92 |
| Class Behavior match | 45% | 52% | 53% |

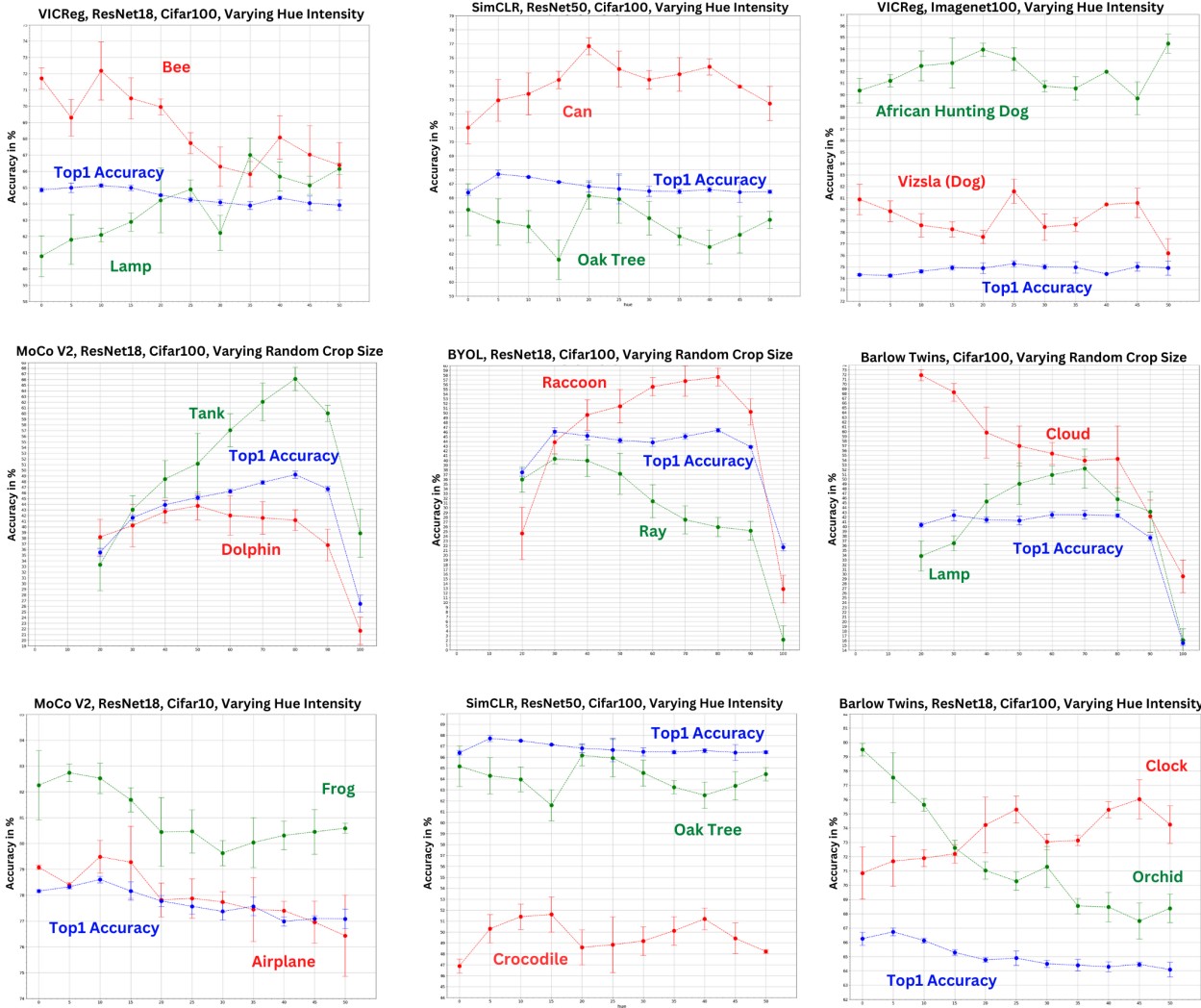

Figure 11: Inter-class accuracy results for Resnet architectures trained with various SSRL methods on the benchmark datasets Cifar10, Cifar100 and Imagenet100, as the parameters of different image transformations are varied. Each dot and associated error bar reflects the mean and standard deviation of five runs for Cifar, each with a different random seed. The results demonstrate that while overall accuracy remains relatively consistent across a range of transformation parameters, these transformations can have a subtle but significant impact on individual class performance, either favoring or penalizing specific classes.

Table 8: The outcomes of linear evaluation for various architectures (VGG11, ResNet18, ConvNeXt-Tiny) trained with different self-supervised representation learning (SSRL) methods (MoCov2 Chen et al. (2020b), BYOL Grill et al. (2020)) on the MNIST dataset LeCun et al. (1998). The models were trained using a set of transformations consisting of random rotations, crops, flips, and random erasing. Notably, the results exhibit consistent patterns across the different self-supervised approaches and backbone architectures, demonstrating the robustness of the observed outcomes.

| METHOD | VGG11 | RESNET18 | CONVNEXT-TINY |
|--------|-------|----------|---------------|
| BYOL   | 61.3  | 62.5     | 51.6          |
| MOCOV2 | 62.1  | 63.8     | 58.7          |

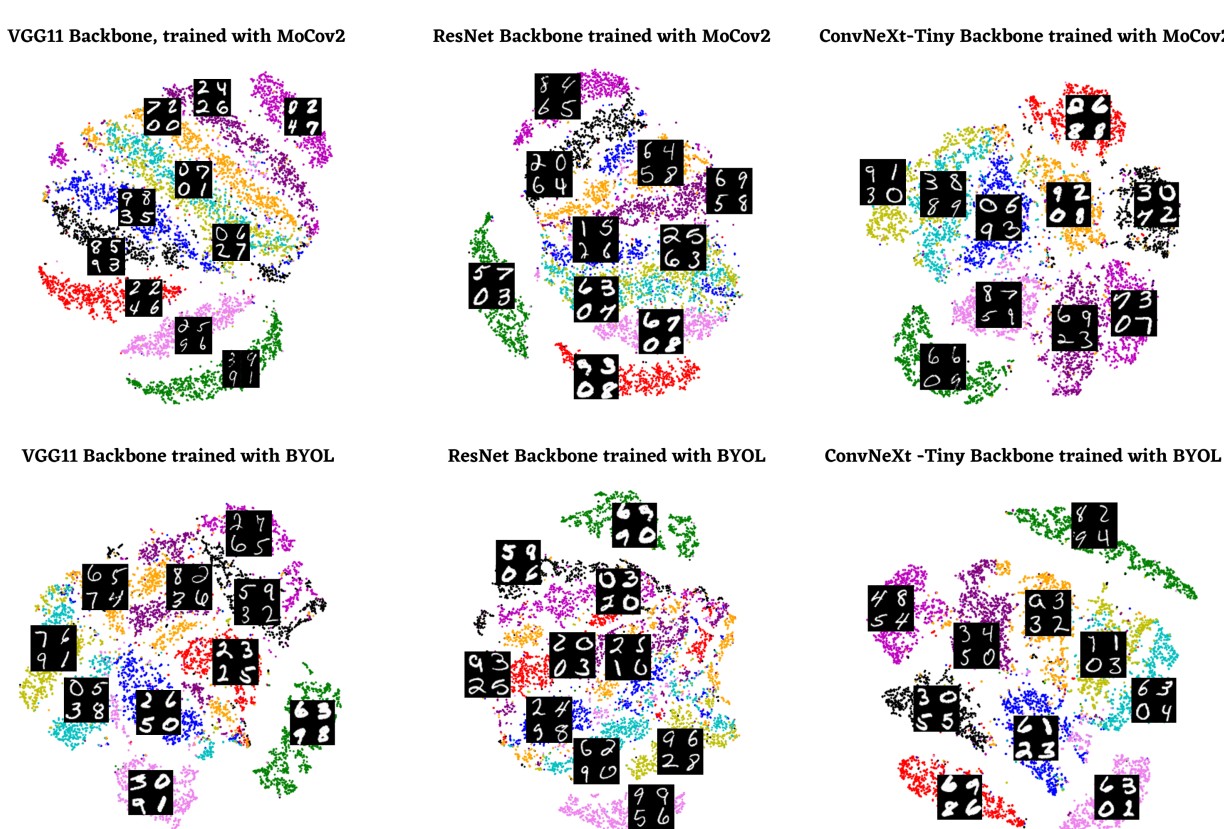

Figure 12: Clustering results of MNIST dataset using various backbones trained with BYOL and MoCov2 as SSRL approaches, using specific image transformations that preserve the handwriting style and line thickness.

