# OpenReview forum: "No Free Lunch in Self Supervised Representation Learning"
_TMLR — Rejected by TMLR_

### Review · Reviewer_hFs3 · 2023-07-04

**Summary Of Contributions:**

Self-supervised learning in computer vision depends on specific image transformations to derive meaningful features, yet the effects of these transformations haven't been extensively examined. This study delves into this aspect, revealing that each category in a dataset can be differently influenced in a manageable way by the transformations. Furthermore, in areas like microscopy imagery, where class differences are more nuanced than in natural images, these transformations significantly impact feature encoding. Ultimately, the study suggests that strategic transformation selection, depending on the desired features, can serve as a form of supervision, thus enhancing performance and representation quality.

**Audience:**

Yes

**Broader Impact Concerns:**

This work, although being empirical, has its own interest. Most of the time people either blindly or automatically choose augmentation hyperparameters, or people do not have such kind of supervision signal to choose augmentations.

This work will inspire the Machine Learning community to study principles in augmentations vs. subclasses.

**Claims And Evidence:**

Yes

**Requested Changes:**

Some further questions and concerns:

- CIFAR-10/-100 is still a small-scale benchmark. WordNet and its hierarchy [1] could be useful for a more principled study on the preference of transformations of different classes.
- Are controlled (sub/super)classes shared by different SSL methods and backbones?
- Is it possible to first quantify properties of images of different classes (intrinsic dimension, spectrum of feature covariance, etc.) and then find possible correlations of these abstracted properties vs. the preferred transformations of each class?

[1] “I Am Going MAD: Maximum Discrepancy Competition for Comparing Classifiers Adaptively” Wang et al. 2020

**Strengths And Weaknesses:**

Strengths
+ Conduct comprehensive experiments on the effect of different transformations on class-wise distinguishability and clustering performance.
+ Raise attention to special domains (e.g. medical image) where subtle transformations can already make classes undistinguishable. In this situation, supervision signals may be necessary to choose appropriate transformations; however, this kind of supervision is prohibitive in SSL.

Weaknesses
- This work is a bit too empirical and did not provide meaningful suggestions/guidance/algorithms to readers to adopt findings in their own experiments.
- Some subtitles are too long and thus are less informative, like those in section 3.

---

> ### Author Response · Authors · 2023-08-03
> **Response to Reviewer hFs3**
>
> We would like to sincerely thank all reviewers for their valuable responses to our submission. We are glad that the wide range of studies and the extension beyond natural images caught the interest of the reviewers. We understand however that some specific weaknesses appear in our submission, toward which we will respond in the following response. We have submitted a new revision of the text with all the changes described below, as well as with changes requested by other fellow reviewers, as summarized in our comment above.
>
> ### Empirical nature of the work :
>
> We concur with the reviewer's assessment regarding the empirical nature of our work. We strongly believe that the issue of transformation choice demands more attention from the community, as it is critical to develop a better understanding of the impact of transformations and propose more tailored approaches. Currently, transformation parameters are often selected randomly. We hope that our work will help quantify the effects of these transformations on various levels, sparking new lines of inspiration and providing a guiding framework for future work on transformation choice principles.
>
> ### Subtitle Length :
>
> We thank the reviewer for this observation, which we have taken into account in the newly submitted manuscript, and have made sure to make shorter and more informative titles.
>
> ### Study on WordNet and its hierarchy :
>
> We agree with the reviewer that experiments on datasets like WordNet and its hierarchy could provide invaluable insights into the transformation preferences of different classes, thus contributing to a more principled study. However, we believe that an investigation of this scale and depth merits its own standalone work, rather than being integrated into our current manuscript. Our current work primarily aims to provide an initial, broad quantification of the effects of transformation choice on the training of different Self-Supervised models. We believe this will pave the way for more focused research exploring the nuances of each aspect discussed in our work.
>
> ### Shared controlled (sub/super)classes :
>
> We conducted an analysis to determine which classes are shared across different SSL methods. The results are summarized in the following table. For this analysis, we used Barlow Twins, BYOL, MoCo v2, SimCLR, and VICReg models trained with a ResNet18 on Cifar100.
>
> | Number of shared classes with negative correlations | Hue Intensity | Color Jitter Probability | Crop size |
> |---|---|---|---|
> | In all 5 SSL approaches | 51 | 0 | 0 |
> | In a minimum of 3 SSL approaches | 97 | 3 | 4 |
> | In a minimum of 2 SSL approaches | 99 | 27 | 8 |
>
> By comparing the number of shared classes with interclass bias in at least two SSL methods, with the number of classes with interclass bias per SSL approach in Figure 2(a) of our manuscript, we observed a significant overlap among the classes with interclass bias across models. However, apart from the case of Hue Intensity, there are no classes that consistently show interclass bias across all five SSL methods. This suggests that the impact of transformation choice on classes might vary depending on the SSL approach selected for training.

---

> > ### Author Response · Authors · 2023-08-03
> > **Response to reviewer's concerns**
> >
> > ### Correlations of abstract class properties versus preferred transformations :
> >
> > We appreciate the reviewer's suggestion about establishing correlations between the preferred transformations for each class and quantified properties of the classes. This would indeed allow us to offer meaningful advice for readers to apply in their own experiments. Therefore, we conducted additional experiments to understand the relationship between a class's accuracy behavior when a chosen transformation is varied and calculated properties for each class in Cifar100.
> > To determine if a class's accuracy is ascending, descending, or random, relative to variations in a transformation value, we first compute the slope of the linear regression line that best fits the accuracy-transformation data for each class and each model. If the slope is significantly positive, we categorize the accuracy as ascending with the transformation variations. If it is significantly negative, it is descending, and if it is close to zero, it is random.
> > As for quantifying class properties, we compute the Intrinsic Dimension and Spectrum of Feature Covariance for each Cifar100 class, using the features of a ResNet model pretrained in a supervised manner on ImageNet. Additionally, we calculate Texture Analysis and Fourier Transform measures. Using ANOVA (and MANOVA) correlation metrics, we then compute the correlation between these measures and the behavior of each class while varying a specific transformation.
> > The following table summarizes our results. Correlation values significantly larger than 1 suggest that the variation between the behavior groups is larger than the variation within each behavior group. This indicates a significant difference between at least two of the behavior groups in relation to the variation of the chosen transformation. All correlation values have a p-value < 0.05, except for those denoted with an asterisk (*).
> >
> >
> > | Class Properties | Hue Intensity | Color Jitter Probability | Crop Size |
> > |---|---|---|---|
> > | Intrinsic Dimension | 16.64 | 19.15 | 0.21* |
> > | Texture Analysis | 16.39 | 4.89 | 0.18* |
> > | Fourrier Transform | 0.71* | 7 | 5.19 |
> > | Spectrum of Feature Covariance | 1.27 | 0.56* | 0.97* |

---

### Review · Reviewer_fZou · 2023-07-14

**Summary Of Contributions:**

- The paper presents an analysis of data augmentation/transformations for self-supervised learning on performance of various tasks.
- Various experiments are carried out to show how various per-class results are correlated with the choice and combination of transformations.
- An analysis of such augmentations for microscopy datasets is performed where distinction between classes is often unclear and subtle.

**Audience:**

Yes

**Broader Impact Concerns:**

While I do not see any obvious negative ethical implications, authors are urged to include a section on broader impact concerns.

**Claims And Evidence:**

Yes

**Requested Changes:**

Please refer to the weaknesses section.
Questions :
- Did the authors explore linear evaluation vs finetuning of features ?
- I am not sure why the phenomena is termed as a inter-class 'bias'.
- Table 2, unclear why was ResNet 101 used for this table while First set and Second set seems to use VGG ? "Pretrained ResNet-101" - is this pre-trained with labels ?

Suggestions:
- Parts of the paper are ambiguous. For example :
    - Figure 2(b). Unclear interpretation of this figure. Captions are not self-sufficient.
    - Page 6 : "The first set, .. " make this choice of set of augmentations and the reasoning more clear in the next draft.
    - Some sentences are too long : For example : first sentence of 3.3
- Provide a brief intuition of the AMI and silhouette score in the main paper for completeness.


Minor suggestions:
- Writing :
    - Introduction, Line 1 : reword it since not all SSRL approaches need this objective
- It would be good to discuss another relevant line of works which obtain additional positives or negatives in the feature space instead of augmenting in the input.

[a] Hard Negative Mixing for Contrastive Learning
[b] HaLP: Hallucinating Latent Positives for Skeleton-based Self-Supervised Learning of Actions

**Strengths And Weaknesses:**

Strengths:
- Data augmentations have been shown to be the key in most self-supervised learning algorithms. The paper further justifies this observation by performing interesting experiments on effect of choice and combination of such augmentations on various tasks.
- The experimental setup and analysis particularly for microscopy images might be of interest to researchers working in that specific area.

Weaknesses:
- It seems like the results presented in the paper are for pretraining and evaluation on the same dataset. Authors should make this more clear.
- This is a disadvantage of this work since the key use of self-supervised learning is to be able to use pre-trained models to transfer to other datasets. What do the authors think about this. Would such an analysis be feasible for transfer learning ? Would you see consistent trends ?
- The clustering and representation information sections is interesting but I wonder if such technique would work for more realistic datasets unlike MNIST.
- By carefully choosing transformation based on performance on the same downstream task cannot be generalized since this indirectly uses label information. What do the authors think about this aspect?

---

> ### Author Response · Authors · 2023-08-03
> **Response to Reviewer fZou**
>
> We would like to sincerely thank all reviewers for their valuable responses to our submission. We are glad that the wide range of studies and the extension beyond natural images caught the interest of the reviewers. We understand however that some specific weaknesses appear in our submission, toward which we will respond in the following response. We have submitted a new revision of the text with all the changes described below, as well as with changes requested by other fellow reviewers, which we summarise in one comment above.
>
> ### Pretraining, evaluation and transfer learning :
>
> We acknowledge the reviewer's confusion regarding the execution of pretraining and evaluation on the same dataset, as well as their curiosity about the feasibility of this analysis in transfer learning. The results presented in our manuscript are for pretraining and evaluation performed on the same dataset (with a separation between training and test data, and evaluation conducted on the test data). We conduct a Linear Evaluation (Linear Probing) on the test set, aligning with the commonly used approach in the literature for evaluating self-supervised approaches. This evaluation task assesses the model's ability to learn general representations that are useful for classification in our specific case. Other evaluation tasks can also be used to evaluate the model’s performance in various other downstream tasks.
>
> Given one of the primary objectives of self-supervised learning is to train models without the need for labels, we believe our use of the same dataset (with distinct sets) for training and evaluation presents an advantage. This approach enables us to assess how the representations learned without the use of labels are impacted by the choice of transformations at the class level. Unfortunately, due to the scope of our study being limited to small-to-medium scale datasets, we find our experiments may not provide compelling scientific conclusions in the context of transfer learning, which demands massive datasets within the context of Self-Supervised Learning.
>
> However, we believe that an adapted analysis could yield consistent trends on the subject of transfer learning. Existing literature, such as the work by Balestriero et al. 2022, "The Effects of Regularization and Data Augmentation are Class Dependent", demonstrates that the effects of transformations in a supervised learning context with massive datasets (a context where models are frequently used for transfer learning) consistently impact the performance of classes across different datasets with the same learned biases that appear in the original dataset.
>
> ### Clustering on other datasets :
>
> We understand the reviewer's curiosity regarding the applicability of our conclusions on representation information for more realistic datasets, as opposed to MNIST. We wish to emphasize that our results, depicted in Figures 6 and 7, demonstrate the same technique and conclusion on three different microscopy datasets. Although these datasets were selected as clear and easily interpretable examples for the readers, similar experiments could be conducted on natural image datasets. Here, the outcomes would likely be consistent, albeit more nuanced. For instance, they might indicate better representation of certain class-specific information compared to others, as suggested by the results in Figure 1.
>
> ### Choice of transformation versus evaluation using labels :
>
> We understand the reviewer's concern that choosing transformations based on performance on the same downstream task may not be generalizable as it involves label information. We believe the reviewer may have misinterpreted the objective of the paper's experiments. Specifically, we do not use label information to identify the optimal transformation choice. Rather, our goal is to demonstrate and quantify how the representations learned through self-supervised pre-training (without labels) exhibit subtle interclass biases or varying degrees of information loss. This insight is derived from evaluating the frozen models with the dataset labels in an analysis setting. We hope that this understanding stimulates new methodologies in subsequent works on self-supervised learning that might potentially overcome this bias resulting from transformation choice, thereby achieving truly generalizable representations. We do not endorse choosing transformations based on performance evaluated using label information in a practical setting, as this would contradict the principles of Self-Supervised Learning.

---

> > ### Author Response · Authors · 2023-08-03
> > **Response to reviewer's concerns**
> >
> >
> > ### Linear evaluation versus Finetuning :
> >
> > We agree with the reviewer that this is an intriguing research path, and we have conducted additional experiments to investigate this further. We carried out several self-supervised trainings with different SSL approaches on Cifar100, varying the hue intensity parameter, and compared the number of classes with statistically significant negative correlations (p-value < 0.05) after either performing linear evaluation or fine-tuning. Additionally, we categorized the accuracy trend of each class as ascending, descending, or random according to its slope. We utilized these categories to discern whether the shared negatively correlated classes across the experiments on linear evaluation and finetuning exhibited similar trends. The subsequent table summarizes our findings.
> >
> > | Methodology | Simclr | BYOL | VicReg |
> > |---|---|---|---|
> > | Resnet18 + Linear Evaluation | 97 | 80 | 92 |
> > | Resnet18 + Finetuning | 96 | 100 | 92 |
> > | Class trend matching (three trends/behaviors) | 45% | 52% | 53% |
> >
> > We observed a strong correlation between the number of classes with negative correlations in both fine-tuning and linear evaluation. Furthermore, there was a relatively high degree of alignment between class behaviors and trends, suggesting that approximately half of the shared classes between fine-tuning and linear evaluation demonstrated exactly the same behavior. Although it might intuitively seem that all classes would share the behavior, we surmise that the original data augmentations used in fine-tuning/linear evaluations (which remained consistent in all our trainings) might have influenced the representations of some classes, thereby altering the behaviors of these classes. However, this did not resolve the issue of interclass bias, as it only changed the behavior of the classes while preserving the bias.
> >
> > ### Inter-class “bias” :
> >
> > We understand the reviewer's uncertainty regarding our use of the term "inter-class bias" to describe the phenomenon in Section 3.1. We acknowledge this confusion and would like to point out that, as demonstrated in Figure 2.a, there is a significant number of classes with statistically significant negative correlations. This suggests that while some classes benefit from the transformation choice, the performance of others deteriorates as a result. Therefore, the model's bias tends towards either one group of classes or another, depending on the transformation choice. Hence, we used the term "bias" to describe this phenomenon. However, we remain open to alternative terminologies that might better encapsulate the nuances of this phenomenon.
> >
> > ### ResNet/VGG usage in Microscopic Images tables :
> >
> > We understand the reviewer's curiosity about why we used VGG for training on microscopic images, while we opted for a pretrained ResNet for transfer learning. The confusion arises due to a lack of detailed experiments on our part, for which we apologize. We include additional results in the revised manuscript, comparing the use of pretrained ResNet with pretrained VGG, as well as training the model with both ResNet and VGG as backbones, for both Tables 2 and 3. The pretrained models were trained in a supervised manner on ImageNet. The table below summarizes our findings.
> >
> > | Backbone | Transformation Approach | Nocodazole | Cytochalasin B | Taxol |
> > |---|---|---|---|---|
> > | VGG | First Set (VGG 13) | 0.19 | 0.27 | 0.16 |
> > |  | Second Set (VGG 13) | 0.37 | 0.45 | 0.38 |
> > |  | Combination of sets (VGG 13) | 0.51 | 0.66 | 0.52 |
> > |  | Pretrained Model (VGG16) | 0.34 | 0.55 | 0.36 |
> > | ResNet | First Set (ResNet18) | 0.17 | 0.25 | 0.15 |
> > |  | Second Set (ResNet18) | 0.33 | 0.42 | 0.31 |
> > |  | Combination of sets (ResNet18) | 0.46 | 0.63 | 0.47 |
> > |  | Pretrained Model (Resnet101) | 0.39 | 0.57 | 0.43 |
> >
> > We chose to train from scratch using ResNet18 due to the low scale of our datasets. The selection of VGG16 over other VGG architectures for the pretrained model trained on ImageNet with labels is justified in the supplementary materials of the revised manuscript. VGG architectures seem to outperform ResNet when training from scratch on our small-scale dataset, while pretrained ResNet models surpass the performance of the pretrained VGG model in transfer learning. This led to our initial choice of training a VGG model and using a pretrained ResNet model.

---

> > > ### Author Response · Authors · 2023-08-03
> > > **Response to reviewer's concerns**
> > >
> > > ### Ambiguity :
> > >
> > > We thank the reviewer for notifying us about the ambiguity in some parts of the paper. We revise these sections and explain them in a more detailed manner in the revised manuscript, while making sure to shorten long sentences.
> > >
> > > ### AMI & Silhouette score :
> > >
> > > We agree with the reviewer that providing an intuition of the AMI and Silhouette scores in the main paper would help complete the content of the draft. We add a small section on these scores in the main paper of the revised manuscript.
> > >
> > > ### Writing :
> > >
> > > We agree with the reviewer that the first line of the introduction requires rewording, to better reflect the differences between different SSRL approaches. We improve this part in the revised manuscript.
> > >
> > > ### Relevant related work :
> > >
> > > We agree with the reviewer that the mentioned works present an interesting avenue to learning representations without augmenting the input, which merits to be discussed. We add a discussion on this line of works in the revised manuscript.
> > >
> > > ### Broader Impact Concerns :
> > >
> > > We agree with the reviewer that our work would benefit from a section on Broader Impact Concerns. We add this section in the revised manuscript.

---

### Review · Reviewer_3QBq · 2023-07-21

**Summary Of Contributions:**

The authors provide a study into the role of image transformations in SSRL. They uncover that the choice of transformation can have varied impacts on performance at the class level, with the ability to improve accuracy for some classes at the potential expense of others. in addition, they suggest that careful selection of transformations allows for optimization of models to encode distinct features into the resulting representations. they also show the implications of transformation selection in SSRL within the biological domain (specifically microscopy images of cells), where class distinctions are often subtle.

**Audience:**

Yes

**Broader Impact Concerns:**

none noted.

**Claims And Evidence:**

No

**Requested Changes:**

1. It could be better if more in-depth discussion of the effects between transformations and the end results are discussed. For example, the authors mention that

   > The reported accuracies reveal that a small crop size benefits the Caterpillar class, as it encourages the model to learn features specific to its texture, while the Crocodile class is negatively impacted in the same range.

   but more discussions can benefit the understanding of this further. For example, it does not seem intuitive that why Caterpillar benefits from model learning texture while Crocodile does not. In fact, I personally tend to think the opposite.

2. The results in Table 3 can be more convincing if the authors can show the resnet trained with different SSRL methods and transformations. At this moment, the results seem to be at a scale not often seen at publications of this tier.

3. The paper is written in a way that there many bragging words, such as
    - we aim to clearly and "unequivocally" measure
    - We "systematically" vary the intensity of the hue
    - desired features yields "remarkable" improvements in result quality
    - The above are just a couple of examples, I will recommend the authors to remove all the words quoted and leave whether the contributions are remarkable or not to the readers.


**Strengths And Weaknesses:**

- strengths

   - the paper presents a wide range of interesting studies of transformation in SSRL
   - the paper extends its relevance beyond natural images and dives into the more complex domain of microscopy images


- weakness

   - although the effects of transformations in class-specific performances have not been discussed, the results found are fairly intuitive, and might not be worth a publication report
   - similarly, the extension to clustering is fairly intuitive
   - the experiments on the microscopy images are not so convincing.

---

> ### Author Response · Authors · 2023-08-03
> **Response to Reviewer 3QBq**
>
> We would like to sincerely thank all reviewers for their valuable responses to our submission. We are glad that the wide range of studies and the extension beyond natural images caught the interest of the reviewers. We understand however that some specific weaknesses appear in our submission, toward which we will respond in the following response. We have submitted a new version of the text with all the changes below.
>
> ### Intuitiveness of results :
>
> We acknowledge the reviewer's perspective that the class-specific performance results and the extension to clustering may appear intuitive and not necessarily warrant a publication. Nonetheless, we assert that this work has significant relevance. Often, individuals, apart from those deeply specialized in the field of self-supervised learning, select transformation parameters at random or without fully comprehending the implications of their choices on the results. We contend that the insights we've gathered about these transformations will not be broadly adopted in the machine learning community unless they are disseminated via a scientific, peer-reviewed publication—even if these findings might seem intuitive to a select few experts in the field of self-supervised learning. As no existing work delves into the influence of transformations in SSL, our aim is to quantify these effects. Though they might be intuitive to some, our findings would help disseminate this knowledge throughout the community and encourage further study into the principles behind these transformations.
>
> ### Not convincing experiments on Microscopy Images :
>
> We recognize that the reviewer found our experiments with microscopy images unconvincing. To address this concern, we've conducted additional experiments, the details of which can be found in the fourth point below.
>
> ### In Depth Discussion of the effects between transformations and results :
>
> We concur with the reviewer on the need for a more detailed discussion on the correlation between transformations and the final results. We also acknowledge that our commentary on the crocodile/caterpillar relationship may be confusing, and that the use of "texture" might be misleading. We've understood that a smaller crop size benefits the Caterpillar class, given the uniformity of body parts throughout its body, a characteristic not shared by the Crocodile class. A smaller crop size might capture a part of the crocodile (such as the tail), which could be attributed to other classes (like a snake). In contrast, all parts of a caterpillar are distinctively caterpillar-like. Although we hoped that the term "texture" encapsulated this nuance, we appreciate the correction from the reviewer.
>
> In the revised manuscript, we elaborate on this relationship and provide a more in-depth discussion about the relationship between transformations and end results. We aim to identify correlations between the preferred transformations for each class and quantified properties of the classes. This analysis could provide meaningful suggestions for readers to implement in their own experiments. To that end, we conduct additional experiments to understand the link between the behavior of a class's accuracy toward varying a chosen transformation, and computed properties for each class in Cifar100.
>
> To ascertain whether the accuracy of a class is ascending, descending, or random with respect to variations of a transformation value, we calculate the slope of the linear regression line fitted to the accuracy-transformation data for each class and model. If the slope is significantly positive, we consider the accuracy ascending with transformation variations. Conversely, if it is significantly negative, it denotes a descending trend, and if it is close to zero, we classify it as random.
>
> As for quantifying class properties, we compute the Intrinsic Dimension and the Spectrum of Feature Covariance for each class in Cifar100, using the features of a ResNet pretrained in a supervised manner on ImageNet. We also calculate Texture Analysis and Fourier Transform measures. We then calculate the ANOVA (and MANOVA) correlation metrics between these measures and the behavior of each class while varying a specific transformation.
>
> The following table summarizes our results. Correlation values significantly larger than one suggest that the variation between the behavior groups is larger than the variation within each behavior group. This indicates a significant difference between at least two of the behavior groups concerning the variation of the chosen transformation. All correlation values have a p-value < 0.05, except for those denoted with an (*).
>
> | Class Properties | Hue Intensity | Color Jitter Probability | Crop Size |
> |---|---|---|---|
> | Intrinsic Dimension | 16.64 | 19.15 | 0.21* |
> | Texture Analysis | 16.39 | 4.89 | 0.18* |
> | Fourrier Transform | 0.71* | 7 | 5.19 |
> | Spectrum of Feature Covariance | 1.27 | 0.56* | 0.97* |

---

> > ### Author Response · Authors · 2023-08-03
> > **Response to Reviewer's concerns**
> >
> > ### Results on Table 3 on Microscopic Images :
> >
> > We appreciate the reviewer's feedback regarding Table 3, and their interest in observing the models trained with different Self-Supervised Representation Learning (SSRL) methods. Initially, we would like to clarify that the pre-trained models were trained in a supervised manner on ImageNet. Additionally, we conducted a detailed analysis of the effects of different transformations on the scores, as presented in Figure 8. In the revised manuscript, we have included additional experiments showcasing the results from training with various SSRL approaches. The following table summarizes our supplementary experiments.
> >
> > | Transformations | SSRL approach | Backbone | Nocodazole | Cytochalasin B | Taxol |
> > |---|---|---|---|---|---|
> > | First Set | MoCo v2 | VGG13 | 0.19 | 0.27 | 0.16 |
> > |  |  | ResNet18 | 0.17 | 0.25 | 0.15 |
> > |  | Byol | VGG13 | 0.21 | 0.28 | 0.19 |
> > |  |  | ResNet18 | 0.2 | 0.25 | 0.17 |
> > |  | VICReg | VGG13 | 0.19 | 0.26 | 0.2 |
> > |  |  | ResNet18 | 0.16 | 0.25 | 0.21 |
> > | Second Set | MoCo v2 | VGG13 | 0.37 | 0.45 | 0.38 |
> > |  |  | ResNet18 | 0.33 | 0.42 | 0.31 |
> > |  | Byol | VGG13 | 0.38 | 0.48 | 0.41 |
> > |  |  | ResNet18 | 0.35 | 0.44 | 0.34 |
> > |  | VICReg | VGG13 | 0.38 | 0.44 | 0.36 |
> > |  |  | ResNet18 | 0.34 | 0.43 | 0.3 |
> > | Combination of Sets | MoCo v2 | VGG13 | 0.51 | 0.66 | 0.52 |
> > |  |  | ResNet18 | 0.46 | 0.63 | 0.47 |
> > |  | Byol | VGG13 | 0.51 | 0.64 | 0.54 |
> > |  |  | ResNet18 | 0.47 | 0.61 | 0.48 |
> > |  | VICReg | VGG13 | 0.55 | 0.67 | 0.51 |
> > |  |  | ResNet18 | 0.5 | 0.63 | 0.45 |
> > | Pretrained models with Imagenet |  | VGG16 | 0.34 | 0.55 | 0.36 |
> > |  |  | ResNet101 | 0.39 | 0.57 | 0.43 |
> >
> > The scope of our work being one of low-scale datasets, we opted for ResNet18 for training from scratch instead of larger architectures. The justification for choosing VGG16 over other VGG architectures for the pretrained model trained on ImageNet with labels can be found in the supplementary materials of the revised manuscript. It appears that VGG architectures outperform ResNet in training from scratch on our small-scale dataset, whereas pretrained ResNet models surpass the performance of the pretrained VGG model in transfer learning. This informed our initial choice for comparing the results of a VGG model trained from scratch and using a pretrained ResNet model.
> >
> > ### Writing style :
> >
> > We are thankful for the feedback of the reviewer regarding the language used in the paper. Upon reflection, we concur that some phrases might come across as excessive or overly assertive. We correct this issue in the uploaded revised manuscript, by removing all the quoted words and other bragging words that exist in the draft, in order to maintain a balanced and objective throughout.
> > We hope these responses adequately address your concerns.

---

### Author Response · Authors · 2023-08-03
**Updates to Revised Manuscript**

We would like to sincerely thank all reviewers for their valuable responses to our submission. We are glad that the wide range of studies and the extension beyond natural images caught the interest of the reviewers. We summarize here the modifications made in the uploaded revised manuscript :

- Writing :
    - We remove exaggerating words from the text of the draft.
    - We reword Introduction Line 1 to take into account approaches with a different objective.
    - We shorten sentences that are too long.
    - We add an additional discussion of a relevant line of works in the related works : Approaches that do not augment the input, and obtain additional positives and negatives in the feature space.
    - We shorten the length of subtitles to make them more informative.
    - We make the Interpretation of Figure 2(b) clearer in the caption of the figure.
    - We further explore and discuss the effects between transformations and the class level effects in Section 3.1, while making our conclusions clearer on the Caterpillar/Crocodile case.
    - We discuss and explain further the reasoning behind each set of transformations used in the MNIST clustering in Section 3.2.
    - We provide a brief intuition of the AMI and Silhouette score in the main paper for completeness, in Section 3.2.
    - We add a section on broader impact concerns in the conclusion of the main paper.
- Additional Experiments :
    - We add an additional analysis in the Supplementary materials on whether controlled classes with inter-class bias are shared by different SSRL methods.
    - We add an additional analysis and results in Section 3.1 on the correlations between abstract properties of classes, and the effects of each transformation on these classes.
    - We add in the supplementary materials an analysis of the number of classes with negative correlations while using linear evaluation, compared to the number of classes when using finetuning, with a focus on classes with matching behaviors between the two experiments.
    - We add additional experiments in Table 3 and Table 4 (Table 2 and Table 3 in the old manuscript), on the usage of additional SSRL approaches and backbones to measure the ability to separate biological phenotypes and compounds. We additionally add results from both pretrained VGG and pretrained ResNet models, trained on ImageNet in a supervised manner.
    - We add in the supplementary materials an additional figure, with experiments concluding that a VGG16 trained on ImageNet with supervision outperforms other VGG backbones that are trained similarly.

---

### Decision · Action_Editors · 2023-09-10

**Recommendation:** Reject

**Comment:**

Based on the above comments, I think that the current submission does not meet the acceptance criteria of TMLR. Even though all of three reviewers recommend weak accepts, the paper still needs to address the limitation of small-scale experiments and the lack of practical contribution from empirical observations.
In addition, many sentences are too long and repeated statements. The descriptions on experiments and metrics are unclear.
I recommend the authors to resubmit the paper after a major revision based on these comments.

**Audience:**

The empirical analysis on the inter-class bias of transformations in SSRL can bring somewhat attention to the relevant community, especially researchers in domains taking microscopy images. However, many researchers who make use of recent large-scale models would not have interest in these small-scale experiments even without any transfer result.

**Claims And Evidence:**

This paper claims that the choice of transformation during the self-supervised representation learning (SSRL) can make opposite impacts on the task performances between two different classes, which they call inter-class bias. This not only leads to the importance of transformation choice in SSRL but also provides an opportunity in controlling the transformation parameters in SSRL in order to improve the performance of a certain target task.
Overall, these claims are extensively supported by experimental results. However, all of empirical validations are performed on small datasets with small networks. Moreover, the claims on this inter-class bias from the transformation choice in SSRL seem to be somewhat straightforward in that some transformations in SSRL sometimes can naturally have negative influences on some classes. I think that based on findings more contribution can come from some practical strategies on how we can efficiently control the transformation in SSRL in improving the performances of some or all classes in a given domain or task. But there is no clear suggestion in this regard in the current submission.

**Resubmission Of Major Revision:**

The authors may consider submitting a major revision at a later time.